**EMBO** *reports*

# Report

# Targeting monocytic Occludin impairs transendothelial migration and HIV neuroinvasion

Diana Brychka [1,2], Nilda Vanesa Ayala-Nunez[1,2,8], Amandine Dupas [3,4,5,6], Yonis Bare [1,2], Emma Partiot [1,2], Vincent Mittelheisser[3,4,5,6], Vincent Lucansky[1,2,7], Jacky G Goetz [3,4,5,6], Naël Osmani[3,4,5,6] & Raphael Gaudin [1,2 ✉]

## Abstract

**Transmigration of circulating monocytes from the bloodstream to tissues represents an early hallmark of inflammation. This process plays a pivotal role during viral neuroinvasion, encephalitis, and HIV-associated neurocognitive disorders. How monocytes locally unzip endothelial tight junction-associated proteins (TJAPs), without perturbing impermeability, to reach the central nervous system remains poorly understood. Here, we show that human circulating monocytes express the TJAP Occludin (OCLN) to promote transmigration through endothelial cells. We found that human monocytic OCLN (hmOCLN) clusters at monocyte-endothelium interface, while modulation of hmOCLN expression significantly impacts monocyte transmigration. Furthermore, we designed OCLN-derived peptides targeting its extracellular loops (EL) and show that transmigration of treated monocytes is inhibited in vitro and in zebrafish embryos, while preserving vascular integrity. Monocyte transmigration toward the brain is an important process for HIV neuroinvasion and we found that the OCLN-derived peptides significantly inhibit HIV dissemination to cerebral organoids. In conclusion, our study identifies an important role for monocytic OCLN during transmigration and provides a proof-of-concept for the development of mitigation strategies to prevent monocyte infiltration and viral neuroinvasion.**

**Keywords** Endothelial Cells; AIDS; Central Nervous System; Tight Junction; Zebrafish
**Subject Categories** Cell Adhesion, Polarity & Cytoskeleton; Immunology; Microbiology, Virology & Host Pathogen Interaction

## Introduction

The neurovascular unit (NVU), also referred to as the blood–brain barrier (BBB), is a complex structure, whose integrity is critical to protect the central nervous system (CNS). Passage through the NVU is a tightly regulated process, and monocyte infiltration represents a hallmark of CNS inflammation. Indeed, monocytes are the first immune cells recruited to the CNS and a major contributor of neurological disorders, particularly important in the context of viral encephalitis (Garre and Yang, 2018; Terry et al, 2012). The transmigration of immune cells from the bloodstream toward the CNS is mediated by their adherence to the endothelial cell membrane of the NVU, followed by squeezing of the immune cell in between endothelial cells, and release in the CNS. The ability of monocytes to transmigrate through the NVU is a pioneering event that can tilt the fragile balance between a healthy or a disordered brain. Understanding the molecular mechanisms involved in monocyte transmigration to the brain is thus highly needed to envision novel strategies to modulate neuroinvasion of immune cells and inhibit neurotropic infections.

Monocyte transmigration is a multistep process requiring a number of essential molecular partners at the cell surface as well as dramatic cytoskeleton and membrane rearrangements to cross the endothelium (Gerhardt and Ley, 2015). The NVU exhibits an additional level of complexity, as it possesses the most impermeant endothelium of the human body sealed by specific tight junctions (TJs). Monocytes can cross the NVU either through paracellular transmigration (migration between endothelial cells) or transcellular migration (migration through endothelial cells) (Muller, 2011). For paracellular transmigration, the mechanisms used by monocytes to cross endothelial TJs during transmigration, while avoiding NVU disruption, remain incompletely understood. Endothelial cells composing the NVU are formed of specific TJ-associated proteins (TJAPs), including Claudin-5 (CLDN5) and Occludin (OCLN). These proteins have been designated as potential targets for the prevention of viral neuroinvasion (Gaudin et al, 2022). Hence, a better understanding of the molecular mechanism involved in monocyte transmigration is important to develop new antiviral strategies that would aim at preventing neuroinvasion.

The CNS is considered a major HIV reservoir, whose infection causes HIV-associated neurocognitive disorders (HAND) (Saylor et al, 2016). Evidences designate macrophages as a main HIV reservoir, even

[1]CNRS, Institut de Recherche en Infectiologie de Montpellier (IRIM), Montpellier, France. [2]Univ Montpellier, Montpellier, France. [3]Tumor Biomechanics, INSERM UMR_S1109, Strasbourg, France. [4]Université de Strasbourg, Strasbourg, France. [5]Fédération de Médecine Translationnelle de Strasbourg (FMTS), Strasbourg, France. [6]Equipe Labellisée Ligue Contre le Cancer, Strasbourg, France. [7]Jessenius Faculty of Medicine in Martin (JFMED CU), Department of Pathophysiology, Comenius University in Bratislava, Martin, Slovakia. [8]Present address: Empa - Swiss Federal Laboratories for Materials Science and Technology, Lerchenfeldstrasse 5, 9014 St. Gallen, Switzerland.
✉E-mail: raphael.gaudin@irim.cnrs.fr

under ART (Ganor et al, 2019; Honeycutt et al, 2017; Honeycutt et al, 2016; Igarashi et al, 2001; Wong et al, 2019). The importance of the myeloid lineage as a master reservoir has also been highlighted by the fact that only patients transplanted with CCR5 Δ32/Δ32 haematopoietic stem-cells, non-permissive to "macrophage-tropic" virus, were cured from HIV-1, while patients transplanted with wildtype CCR5 cells experienced viral rebounds after ART interruption (Cummins et al, 2017; Gupta et al, 2019; Henrich et al, 2014). Infected monocyte-derived macrophages have been detected in the brain of patients despite antiretroviral therapy (ART) (Churchill et al, 2006), but the initial establishment of macrophages as HIV reservoirs in the CNS is not a well-understood process. It was previously proposed that HIV takes advantage of blood-circulating monocytes, enhancing transmigration to cross the NVU using a "Trojan horse" strategy (Ayala-Nunez and Gaudin, 2020; Hazleton et al, 2010). Monocytes are able to capture and store infectious viral particles without being infected themselves (Pino et al, 2015), a "hiding" strategy that plays a major role during HIV dissemination to the CNS (Kincer et al, 2022). Although infrequent, several reports from different groups also indicated that circulating monocytes contain integrated HIV-1 DNA, even in individuals under ART (Lambotte et al, 2000; Massanella et al, 2019; Sonza et al, 2001; Zhu, 2002), highlighting the importance of monocytes as a potential source of infected macrophages in the CNS. Therefore, a better understanding of the molecular mechanisms leading to monocyte transmigration is crucial to envision novel strategies to prevent HIV neuroinvasion.

Several TJAPs have been proposed to play a role during leukocyte transmigration (Gerhardt and Ley, 2015; Vestweber, 2015) and here, we reinvestigated the function of the most expressed monocytic TJAPs (mTJAPs) during this process. First, we performed a mini-screen in THP-1 monocytic cells using siRNA targeting the most expressed mTJAP RNAs. OCLN silencing in THP-1 cells significantly decreases the ability of monocytes to transmigrate through the human cerebral microvascular endothelial D3 cell line (hCMEC/D3), while hmOCLN overexpression in human primary monocytes, isolated from healthy blood donors, results in increased transmigration. Furthermore, we find that monocytic OCLN (hmOCLN) accumulates transiently at the interface between monocytes and endothelial cells. To prevent interaction between hmOCLN and endothelial OCLN, we designed short peptides copying regions of the extracellular loop 1 or 2 of OCLN (EL1 and EL2, respectively). We show that the EL2 peptide significantly prevents monocyte transmigration in vitro and in zebrafish, while preserving endothelial impermeability. Finally, we reveal that treatment with the OCLN-derived peptides prevents monocyte infiltration and HIV neuroinvasion in cortical organoids. In conclusion, our work highlights the importance of hmOCLN during monocyte transmigration and proposes a strategy based on the use of OCLN-derived peptides to control the access of monocytes to the CNS, offering innovative mitigation strategies, such as in the context of HIV brain reservoir establishment.

## Results and discussion

### Monocytic Occludin promotes monocyte transmigration through the BBB-like endothelium

To evaluate whether TJAPs are expressed by monocytes, we first interrogated RNAseq databases (Edgar et al, 2002), seeking for TJAP-coding mRNAs in human primary monocytes and the monocytic cell line THP-1 (Fig. 1A). We found that a subset of TJAP-coding mRNAs was expressed by these cells, including JAM-A, previously associated to transmigration (Gerhardt and Ley, 2015). To test if the monocytic expression of TJAPs participates in monocyte transmigration, we knocked-down 21 of the most expressed mTJAP-coding RNAs in THP-1 cells using siRNA and performed a transmigration assay through hCMEC/D3 cells (Weksler et al, 2005) (Fig. 1B). Compared to control, silencing of OCLN resulted in a two-fold decrease of the number of transmigrated monocytes, while silencing of Claudin-23 (CLDN23) also resulted in a significant down-regulation of monocyte transmigration. In contrast, the other silenced genes did not seem to play a significant role in this process, although this cannot be excluded at this stage as knock-down efficiency was not assessed in our initial screen.

Because the literature on CLDN23 is relatively scarce and the phenotype less pronounced than for OCLN, we chose to focus our subsequent study on the role of human monocytic OCLN (hmOCLN) during transmigration through the endothelium. First, we built a lentivector expressing 2 guide RNAs (gRNAs) targeting OCLN, a Cas9-specific tracrRNA and the Cas9 enzyme (Fig. EV1A, and Addgene #208398). This construct also codes for puromycin resistance and the E2-Crimson far-red fluorescent protein to help for the selection of transduced cells. This construct was used to produce lentivectors and transduce THP-1 cells, allowing to obtain a CRISPR edited THP-1 OCLN knock-out (KO) cell line. Western blot analysis of OCLN protein expression in these cells showed no band at the expected size (Fig. EV1B). Of note, a band of very dim intensity and at a lower size was observed, whose nature is unknown, but whether this represents some remaining OCLN isoforms or non-specific staining, its intensity remains at marginal levels compared to control cells. This cell line was the only clone of THP-1 that showed the disappearance of the band expected at the size of OCLN. In order to thereafter compare transmigration from the same clonal cells, THP-1 OCLN KO cells were rescued using an EGFP-OCLN coding vector or an EGFP-OCLN with deletion of the C-terminal tail (EGFP-OCLN-ΔC). The C-terminal tail of OCLN is the cytosolic part of the protein that interacts with ZO-1 and other actin-associated components (Li et al, 2005; Muller et al, 2005). Its removal was shown to retain plasma membrane localization of OCLN, but lacking anastomosing functionality (Kuo et al, 2022). As a control, THP-1 OCLN KO cells were transduced with an irrelevant EGFP-CAAX coding vector, CAAX being the farnelysation motif of the HRas protein known to address it to the plasma membrane (PM; (Manne et al, 1990; Prior and Hancock, 2001)). Analysis of the expression level of these constructs in THP-1 showed that EGFP-OCLN and EGFP-OCLN-ΔC were expressed at similar levels while the EGFP-CAAX construct was more expressed (Fig. EV1C,D). Using these cells, we showed that transmigration of THP-1 EGFP-OCLN expressing cells was significantly increased compared to EGFP-CAAX and EGFP-OCLN-ΔC counterparts (Fig. 1C), further confirming the importance of OCLN during monocyte transmigration.

In primary human monocytes, endogenous hmOCLN was readily observed by western blot (Fig. EV1E). Despite many attempts however, we could not detect the endogenous OCLN expressed by monocytes by immunofluorescence (IF), likely because it is weakly expressed, exhibiting very low signal-to-noise ratio (Fig. EV1F). To further evaluate the role of OCLN in primary

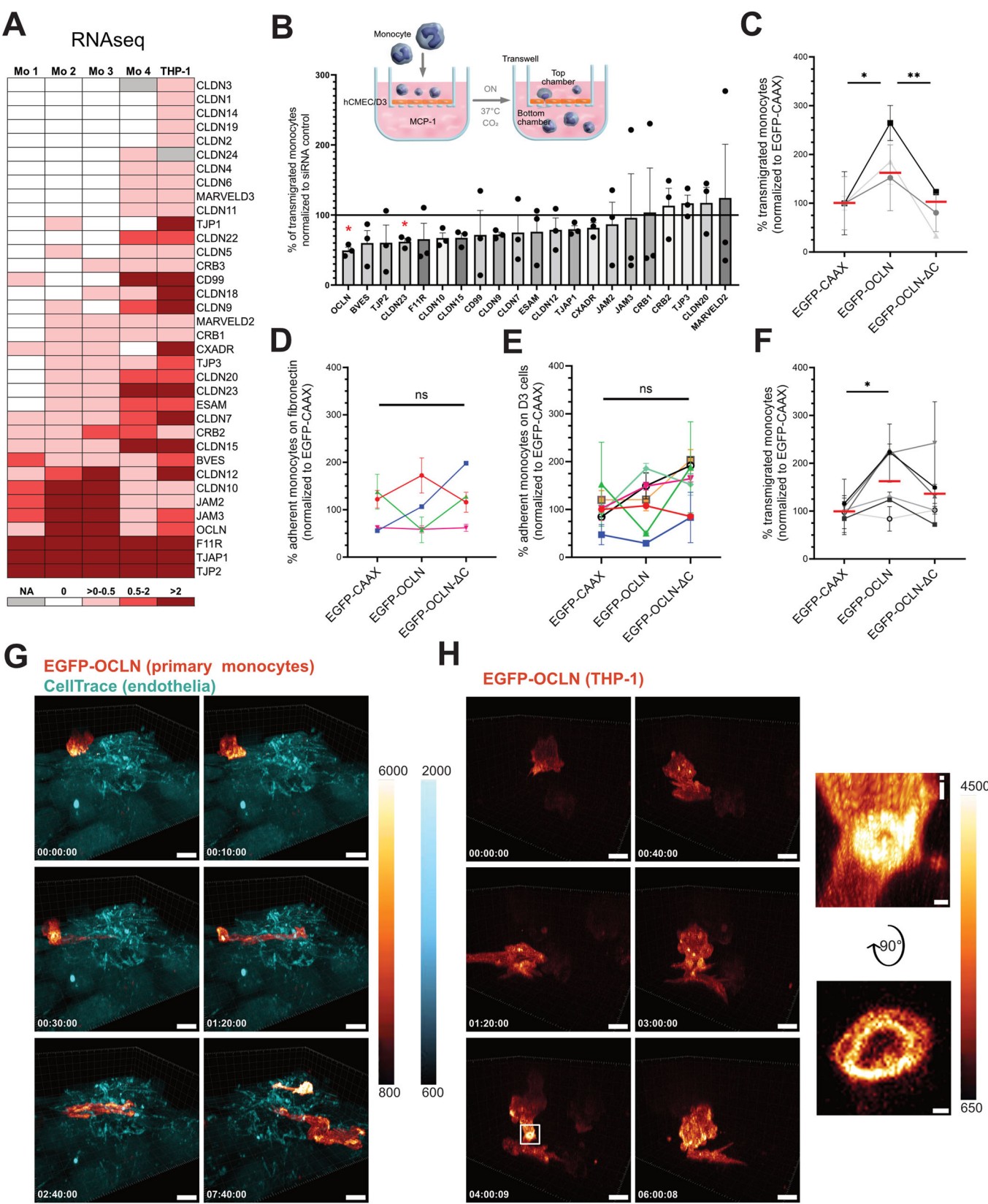

**Figure 1. Monocytic OCLN favors monocyte transmigration across the BBB-like endothelium.**

(A) Heatmap of TJAP-coding mRNAs in human primary monocytes and the THP-1 cell line based on existing RNAseq databases (see Methods). (B) siRNA screen on monocytic TJAP proteins to define their role in transmigration. The scheme represents the experimental setup of a transmigration assay with monocytes crossing a confluent monolayer of brain endothelial cells (hCMEC/D3). TJAPs were selected based on the heatmap shown in A. THP-1 cells were electroporated with the indicated siRNAs and added to the top chamber for 17 h (overnight). The data is presented as percentage of transmigrated THP-1 cells normalized to the siRNA control (black line). The graph shows $n = 3$ independent experiments performed at least in two technical replicates. (C) Percentage of transmigrated OCLN KO THP-1 cells rescued either with EGFP-CAAX, EGFP-OCLN, or EGFP-OCLN-ΔC across hCMEC/D3 monolayer. The data were obtained from $n = 3$ individual experiments performed in two technical replicates normalized to EGFP-CAAX of each experiment. Each symbol's color corresponds to an individual experiment. Red dashes represent the mean of experiments combined together. (D) Percentage of adherent primary monocytes expressing EGFP-CAAX, EGFP-OCLN, or EGFP-OCLN-ΔC on fibronectin substrate measured by flow cytometry. Adhesion time is 30 min. The data were obtained from $n = 4$ monocyte donors. Data were normalized to the average of EGFP-CAAX obtained from each experiment. Each symbol's color corresponds to an individual monocyte donor. (E) Percentage of adherent primary monocytes expressing EGFP-CAAX, EGFP-OCLN, or EGFP-OCLN-ΔC on hCMEC/D3 monolayer measured by flow cytometry. Adhesion time is 30 min. The data represents the mean values obtained from $n = 7$ monocyte donors. Data were normalized to the average of EGFP-CAAX obtained from each experiment. Each symbol's color corresponds to an individual monocyte donor. Error bars are SEM. (F) Percentage of transmigrated primary monocytes expressing EGFP-CAAX, EGFP-OCLN, or EGFP-OCLN-ΔC across hCMEC/D3 monolayer. The data were obtained from $n = 6$ monocyte donors. Data were normalized to the average of EGFP-CAAX obtained from each experiment. Each symbol's color corresponds to an individual monocyte donor. (G, H) 3D time-lapse spinning disk confocal microscopy of transmigrating monocytes. Fluorescence intensity scale is shown on the side of each panel. EGFP is represented using the "red Fire" color-coding, and endothelial cells are stained with a CellTrace marker (cyan) added prior to the addition of the monocytes. Timescales are shown is the bottom right corner and correspond to the time after which imaging acquisition started. (G) Imaging of a primary monocyte transduced with EGFP-OCLN on hCMEC/D3 monolayer. Images were taken every 10 min. Scale bar: 10 μm. Full video can be found in Movie EV1. (H) Imaging of a THP-1 cell transduced with EGFP-OCLN on hCMEC/D3 monolayer (not stained). Images were taken every 10 min. Scale bar: 8 μm. Full video can be found in Movie EV2. The white square highlights OCLN accumulation at the potential interface between hCMEC/D3 cells and the THP-1 cell. (i) Enlarged image from the white square and a 90° flip showing the accumulation of OCLN. Scale bar: 0.7 μm. Data information: In (B, C, F), data are presented as mean ± SEM. In (D, E), error bars represent SEM. Two-tailed Student's t-test $p$ value $< 0.05$ (*), $< 0.01$ (**) or non-significant (ns). Source data are available online for this figure.

cells during transmigration, we optimized a protocol for efficient human monocyte transduction using a Vpx-based strategy (Berger et al, 2009), to overexpress EGFP-CAAX, EGFP-OCLN, or EGFP-OCLN-ΔC. Transduction showed variable efficiency depending on the construct (Fig. EV1G). Transduction was >10% efficiency in monocytes, which is acceptable for primary cell transduction and would not majorly influence the following experiments as we selected only the fluorescent (transduced) cells. The EGFP-OCLN or EGFP-OCLN-ΔC were expressed at similar levels in primary monocytes, while EGFP-CAAX was slightly more expressed (Fig. EV1H).

Analysis of the percentage of transduced primary monocytes that adhered to fibronectin or the hCMEC/D3 endothelium was similar regardless of the construct expressed (Fig. 1D,E). In contrast, overexpression of EGFP-OCLN significantly increased primary monocyte transmigration through hCMEC/D3 compared to EGFP-CAAX-expressing monocytes (Fig. 1F). Of note, our attempts to knock-down/out OCLN in primary monocytes were unsuccessful, which might be due to the relatively stable nature of OCLN, while primary monocytes are kept in culture for only three days maximum before they start to significantly differentiate or die.

Together, our data indicates that the expression of OCLN by monocytes favors transmigration of monocytes through the BBB-like endothelium. Although it may sound unexpected to identify a transmembrane protein known to promote tight cell-cell contacts such as OCLN to be expressed by monocytes, another group previously detected OCLN protein in monocytes, reporting that the protein was overexpressed upon exposure to human cytomegalovirus (Smith et al, 2004). This observation was associated with increased infected monocyte migration, although they did not provide experimental evidence for a link between OCLN and migration at the time. Interestingly, OCLN was shown to regulate the directional migration of epithelial cells (Du et al, 2010). The authors indicated that OCLN would act as an actin scaffold protein at the leading edge of migrating cells. Furthermore, OCLN expressed by γδ T lymphocytes was also proposed to participate

in their intraepithelial migration in mice (Edelblum et al, 2012). These findings reinforce the importance of OCLN during migration of various cell types in tissues, and our study is the first to show that human OCLN plays a role in blood-to-tissue transmigration. Of note, the hCMEC/D3 endothelial cells have tighter TJs than most other cell lines (Weksler et al, 2013), but their TJ tightness does not reach the strength of in vivo NVU. These cells however can become more impermeable upon application of a pulsatile flow, reminiscent of the bloodstream (Weksler et al, 2013), and future work should further investigate the role of OCLN during monocyte transmigration through the NVU under flow conditions.

## Distribution of monocytic OCLN during transmigration

To better understand the dynamics of hmOCLN distribution during monocyte transmigration through hCMEC/D3 cells, we optimized imaging conditions to support 5D live cell microscopy (X, Y, Z, Time, Channels). In the transmigration events caught, we could reproducibly see the following events: upon early steps of monocyte attachment to the endothelial monolayer, hmOCLN was mostly found at the PM of monocytes, with some OCLN polarizing toward the endothelium (Fig. 1G,H, Movie EV1 and Movie EV2). During the transmigration process, OCLN clusters at the monocyte-endothelium interface, while the monocytes started to protrude across. At this stage, we observed the formation of OCLN-containing internal compartments in the monocytes. The OCLN-containing compartments forming during transmigration were not classical recycling endosomes (Rab11- and Rab13-negative; Fig. EV2A,B) nor degradative compartments (Rab7-negative; Fig. EV2C). Of note, the small GTPase Rab13 was previously shown to recycle OCLN (Morimoto et al, 2005), but we could not monitor significant co-localization between Rab13 and OCLN in human monocytes, neither at steady-state nor during transmigration. The OCLN-containing compartments were truly internal (as opposed to deep PM invagination) as fluorescent Dextran, a fluid phase marker, could not access it (Fig. EV2D), indicating that it

likely represents a bona fide intracellular vesicle. Further investigations are needed to determine the nature and function of this compartment during monocyte transmigration.

Toward the end of the transmigration process, the monocyte's whole cell body squeezed in between endothelial cells and flattened beneath the endothelium, in close contact with the coverslip. At these later stages, OCLN was distributed at the PM and in clusters that may be internal. One of the most striking observation of the transmigration process is that OCLN transiently relocalized to inter-endothelial junctions (Fig. 1G,H). In particular, we caught a very transient phenomenon, where an OCLN ring formed at the monocyte-endothelium interface (Fig. 1H right panels, and Movie EV2), which had faster kinetics than our imaging time interval (10 min). Such a structure was never observed in EGFP-CAAX-expressing monocytes (Fig. EV3A and Movie EV3). Finally, we observed that monocytes transduced with EGFP-OCLN-ΔC were not protruding through the endothelium, although the protein also polarized toward the endothelial monolayer (Fig. EV3B and Movie EV4), suggesting that the C-terminal domain of hmOCLN is required to initiate the formation of protrusions and/or the opening of endothelial cell junctions. Although automated quantification could not be made, we report that live imaging of complete transmigration events occurred in 12/35 cells (35%) for EGFP-OCLN, 11/38 cells (29%) for EGFP-CAAX, and 9/40 cells (22%) for EGFP-OCLN-ΔC (data obtained from 4 donors).

Several adhesion molecules (integrins, selectins, immunoglobulin-like superfamily) and chemokine receptors have been involved in monocyte transmigration (Gerhardt and Ley, 2015). However, subcellular insights of the transmigration process are relatively scarce. A seminal study from 1998 investigated the distribution of F-actin and β-catenin in transmigrating rat monocytes (Sandig et al, 1999), and the advent of intravital imaging recently allowed further understanding of this dynamic process (McArdle et al, 2015). However, this later strategy deals with non-human monocytes, has low spatiotemporal resolution and could not readily provide access to the molecular mechanistic associated to this event. Our imaging analyses revealed that hmOCLN can form a ring at monocyte-endothelial contact sites. Such imaging is very challenging and we could not reliably quantify and time this event, although we could reproducibly observe hmOCLN polarization towards the endothelium, illustrating the specific and transient redistribution of hmOCLN during transmigration. Dendritic cells were shown to send dendrites outside the gut and airway epithelia, while expressing OCLN (Blank et al, 2011; Rescigno et al, 2001), but no functional analyses were performed at the time and we showed here that silencing or overexpressing hmOCLN modulates monocyte transmigration through endothelial cells. We hypothesize that hmOCLN can form homotypic interactions with endothelial OCLN in order to squeeze through endothelial TJs while preserving endothelial permeability, a mechanism already suggested for JAM proteins (Gerhardt and Ley, 2015). Although this is an attractive working model, further investigations are required to determine whether these OCLN-mediated intercellular interactions are sufficient to locally retain full impermeability. Moreover, we observed mostly paracellular transmigration, i.e. migration of the monocyte between endothelial cells, but it remains unclear at this stage whether OCLN could also be involved in transcellular migration, a process during which monocytes would migrate through an endothelial cell (Muller, 2011).

## An OCLN-derived peptide inhibits monocyte transmigration in vitro

Peptides derived from the extracellular loops (EL) of OCLN were shown to bind to OCLN, indicating that transcellular homotypic OCLN interactions between cells occur (Chung et al, 2001; Everett et al, 2006; Lacaz-Vieira et al, 1999; Nusrat et al, 2005; Tavelin et al, 2003; Wong and Gumbiner, 1997). In order to prevent the potential interactions of hmOCLN with endothelial OCLN, we designed peptides mimicking the extracellular loop 1 (EL1) or 2 (EL2) of the human OCLN and associated scramble controls (scrEL1 and scrEL2, respectively; Fig. 2A) based on previous literature ((Chung et al, 2001; Everett et al, 2006; Lacaz-Vieira et al, 1999; Nusrat et al, 2005; Tavelin et al, 2003; Wong and Gumbiner, 1997); see Methods for details). Previous work showed that two peptides derived from xenopus OCLN altered epithelial impermeability (Wong and Gumbiner, 1997). Another study showed that a peptide targeting the EL2 of rat OCLN but not EL1, perturbed the endothelial blood-testis barrier in a reversible manner (Chung et al, 2001). This difference could be attributed either to the difference of the peptide sequences, the different cell types investigated, and/or the different species used. To check for the potential of peptides to alter endothelial integrity in our assay, hCMEC/D3 cells were incubated with peptides overnight, and the fluorescent fluid phase marker lucifer yellow was added for 2 additional hours, while the peptides were not washed away. In this assay, none of the peptides significantly affected hCMEC/D3 permeability (Fig. 2B). Moreover, none of the peptides significantly perturbed monocyte adhesion on fibronectin (Fig. 2C). Strikingly, the treatment of human primary monocytes with the EL2 peptide most significantly decreased their transmigration through hCMEC/D3 cells, while EL1 peptide did not (Fig. 2D). The other peptides also showed some anti-transmigratory activity, but it remained less pronounced that for the EL2 peptide. Of note, we used in this assay a second scrEL2 peptide (scrEL2 b) to confirm that the EL2 peptide was specifically more active than its scramble counterparts.

Together, this data suggests that EL2 peptide could represent an interesting strategy to target OCLN-mediated monocyte transmigration. Antibodies targeting the extracellular domain of OCLN have been developed, as they represent an interesting strategy to block Hepatitis C virus (HCV) entry in hepatocytes (Shimizu et al, 2019). Indeed OCLN is an essential entry factor for HCV (Ploss et al, 2009), which drives the dynamics of viral particle internalization (Deffieu et al, 2022). Unfortunately, we did not manage to get access to these proprietary antibodies targeting the ELs of OCLN.

## The OCLN-derived EL2 peptide is not toxic while inhibiting monocyte transmigration in vivo

We next aimed to evaluate the potency of the OCLN-derived peptides to inhibit human monocyte transmigration. To this end, we took advantage of a method that we previously established (Ayala-Nunez et al, 2019), consisting of injecting pre-labeled human monocytes in zebrafish embryos expressing a fluorescent endothelium (Tg(fli1a:egfp-CAAX)); (Fig. 3A,B). This model allows efficient tracking of human monocytes, including virus-infected ones (Ayala-Nunez et al, 2019), transmigrating across endothelial cells of the vascular caudal plexus. Zebrafish has two OCLN

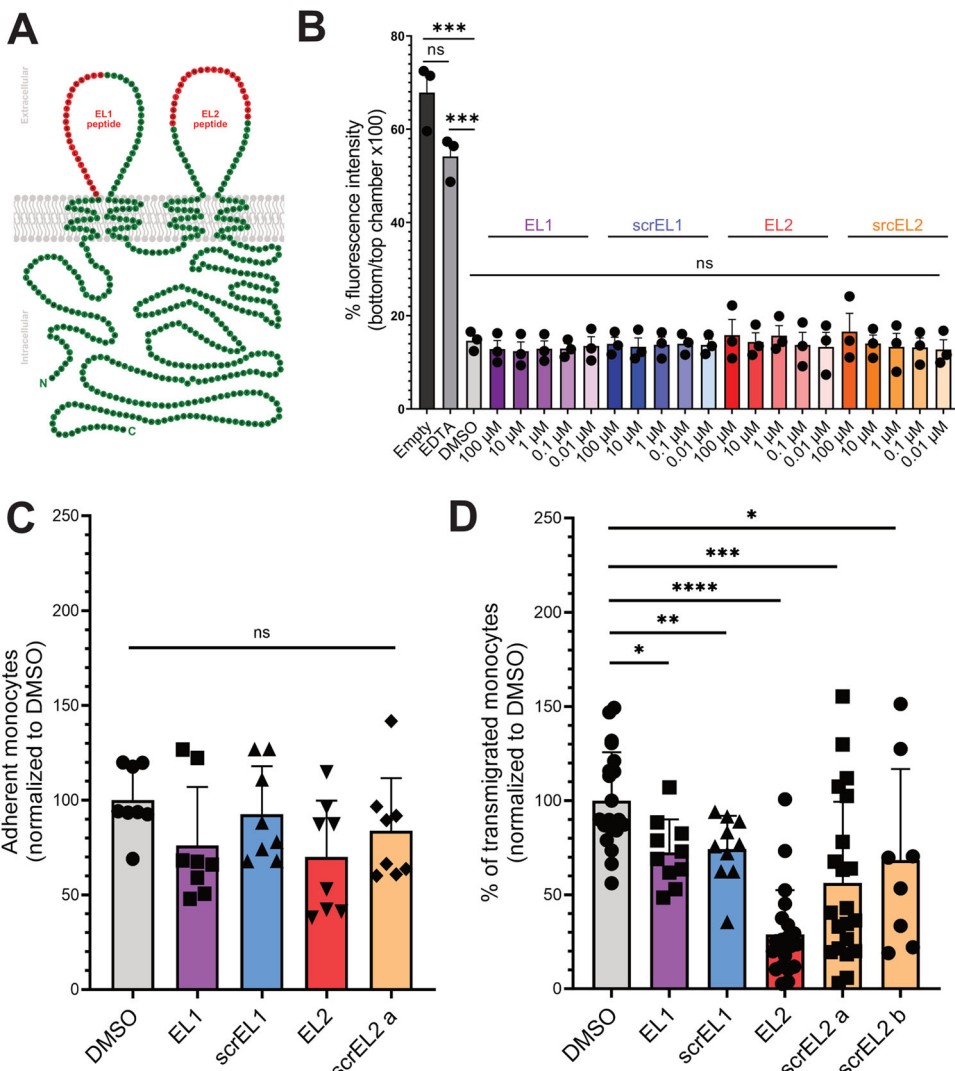

**Figure 2. OCLN-derived EL2 peptide inhibits monocyte transmigration across the BBB-like endothelium.**

(A) Schematic representation of OCLN (green) at the PM with OCLN-derived peptides highlighted in red (see sequence in Methods). (B) Permeability of hCMEC/D3 monolayer after overnight exposure of the monolayer to OCLN-derived peptides or their scramble controls. The data were obtained from $n = 3$ individual experiments. (C) Adhesion of primary monocytes on fibronectin after pre-incubation with OCLN-derived peptides or their scramble controls for 1 h measured with CellTiter Glo. Adhesion time is 30 min. The data were obtained from $n = 4$ donors in 2 individual experiments. Data were normalized to DMSO for each experiment. (D) Percentage of primary monocytes after 1 h exposure to OCLN-derived peptides or their scramble controls that transmigrated across the hCMEC/D3 monolayer overnight. The data were obtained from $n = 8$ individual experiments using 13 donors and normalized to the average DMSO for each experiment. Data information: In (B–D), data are presented as mean ± SEM. Two-tailed Student's t-test $p$ value < 0.05 (*), <0.01 (**), <0.001 (***) or non-significant (ns). Source data are available online for this figure.

paralogues: *oclna* and *oclnb*. Previous RNAseq analysis of the endothelium of zebrafish embryos showed that *oclna* was the most abundant transcript compared to *oclnb* (Fig. EV4A; data extracted from {Bonkhofer, 2019 #1817}). Human OCLN has a 95% consensus sequence with zebrafish OCLN (zOCLN) from the *oclna* isoforom as calculated upon TCoffee alignment ((Chang et al, 2012); Fig. EV4B). The EL1 and EL2 peptide derived from human OCLN had partial, but significant similarities (Fig. EV4C). First, we tested for toxicity and/or vascular permeability in zebrafish embryos by injecting solely the EL1 or EL2 peptides. We did not observe developmental defects as estimated by the morphology and heartbeat of the embryos, and found no significant leakage of the vascular endothelium between the conditions, except for the LPS

control condition (Fig. EV4D–G; (Philip et al, 2017)). Next, human primary monocytes from four individual blood donors were intravenously (i.v.) injected in 48 h post-fertilization (hpf) zebrafish embryos with DMSO, EL1, or EL2 peptides and confocal imaging of the vascular caudal plexus was performed at 6–8 h post-injection (Fig. 3C). Interestingly, the human OCLN-derived EL2 peptide significantly decreased monocyte transmigration, despite relatively high inter-donor variability (Fig. 3D), as we observed in vitro (Fig. 2D).

Together, this data suggests that the EL2 peptide is a potent inhibitor of monocyte transmigration that retains activity in vivo, while not exhibiting noticeable toxicity. As a proof-of-concept, our data highlights for the first time that targeting OCLN in vivo can

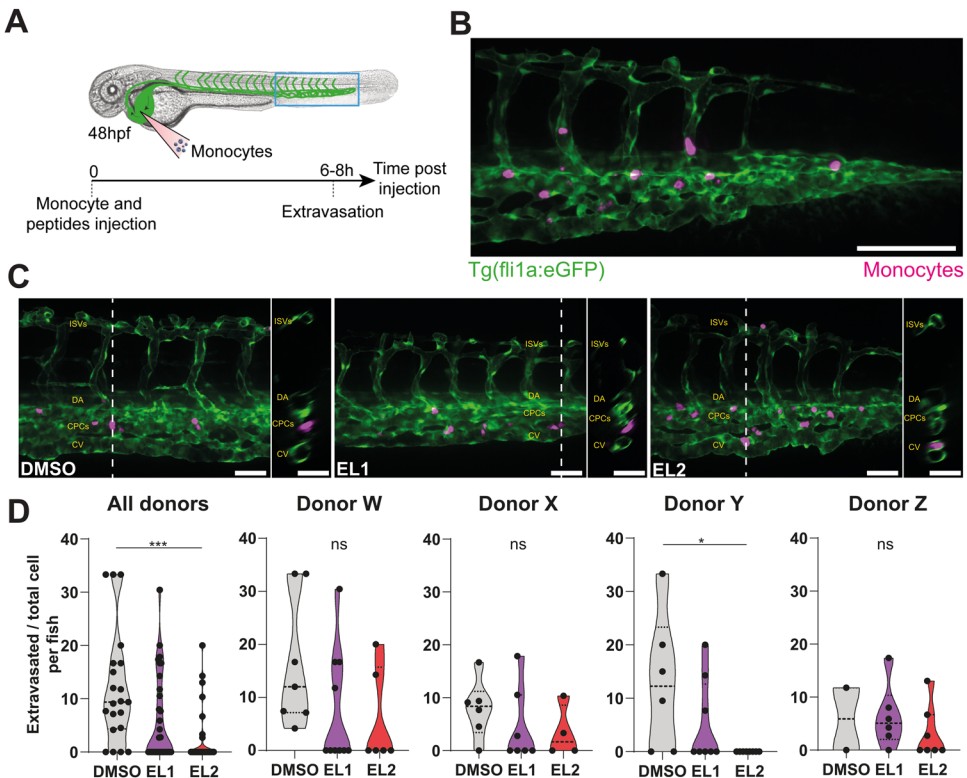

**Figure 3. OCLN-derived EL2 peptide inhibits human monocyte transmigration in vivo.**

(A) Representative scheme of the experimental design associated with the zebrafish model. Human primary monocytes were labeled with CellTrace, and injected into the duct of Cuvier of Tg(fli1a:eGFP-CAAX) zebrafish embryos (GFP-labeled endothelial cells). (B) Zebrafish imaging was done at 6–8 h post injection by spinning-disk confocal microscopy. Representative z-projection of confocal images of monocytes from patient X arrested in the vasculature at the tail of a zebrafish embryo at 6 h post injection Scale bar: 100 µm. (C) Representative images and orthoslice of monocytes at 6 hpi. Scale bar: 50 µm. (D) Violin plots of the ratio of extravasated monocytes from four donors (merged and seperated) at 6–8 h post injection. Each dot represents the ratio calculated from all the monocytes tracked within a fish. The data were obtained from $n = 4$ individual experiments using four different monocyte donors with nDMSO = 21, nEL1 = 31, nEL2 = 25 in total (nDMSO = 7, nEL1 = 10, nEL2 = 6 for donor W; nDMSO = 6, nEL1 = 7, nEL2 = 4 for donor X; nDMSO = 6, nEL1 = 8, nEL2 = 8 for donor Y; nDMSO = 2, nEL1 = 6, nEL2 = 7 for donor Z). P value < 0.05 (*), and <0.0001 (***). Data information: In (D), data are presented as violin plots showing min/max values and quartiles (dotted lines). Statistical difference of non-Gaussian datasets was analyzed using Kruskal–Wallis test with Dunn's post hoc test *p* value < 0.05 (*), < 0.001 (***) or non-significant (ns). Source data are available online for this figure.

represent an attractive approach to modulate monocyte infiltration. Further immunological studies should provide more information as of whether an EL2 peptide treatment could attenuate local tissue inflammation.

## Monocytes carrying HIV-1 particles exhibit increased transmigration

Finally, we wanted to test whether the EL2 peptide could prove to be useful in a human pathological context, using HIV-1 as a model system. HIV-1 has been shown to associate with monocytes and invades the brain using the "Trojan horse" strategy, consisting of being carried through the BBB by hiding in transmigrating circulating cells (Ayala-Nunez and Gaudin, 2020). Here, we showed in our model that the exposure of primary monocytes to HIV-1 promotes monocyte transmigration (Fig. 4A), as previously reported (Williams et al, 2012). While monocyte-derived macrophages and microglia are permissive targets allowing HIV-1 productive infection, circulating monocytes are poorly permissive to HIV-1 infection (Massanella et al, 2019). Nevertheless, monocytes can capture and store infectious viral particles without

being infected themselves (Pino et al, 2015). Here, we show that primary monocytes exposed to fluorescent HIV-1 Gag-iGFP R5 particles are highly positive for EGFP, regardless of antiviral AZT treatment (Fig. EV5A), indicating that the virus particles are carried by monocytes but do not (or very poorly) replicate in them. At the subcellular level, monocytes exposed to Gag-iGFP fluorescent particles exhibit EGFP puncta positive for anti-Gag p17 antibody that remain in AZT-treated monocytes (Fig. EV5B middle panels). Using a VSV-G pseudotyped virus to force productive infection, we also observed the dotted structures, but also a diffuse staining, suggestive of neosynthesis of Gag in the cytosol, which was not observed in AZT-treated cells (Fig. EV5B, right panels).

To test for the importance of monocytes as carriers of HIV particles through the endothelial barrier, hCMEC/D3 cells in transwells were exposed to cell-free HIV-1 particles or monocyte-loaded HIV particles. Unambiguously, monocytes proved to be an exquisite vessel for HIV-1 particles, transporting the virus through the endothelium about a million times more efficiently than cell-free virus (Fig. 4B), highlighting the importance of the transmigration process for HIV neuroinvasion.

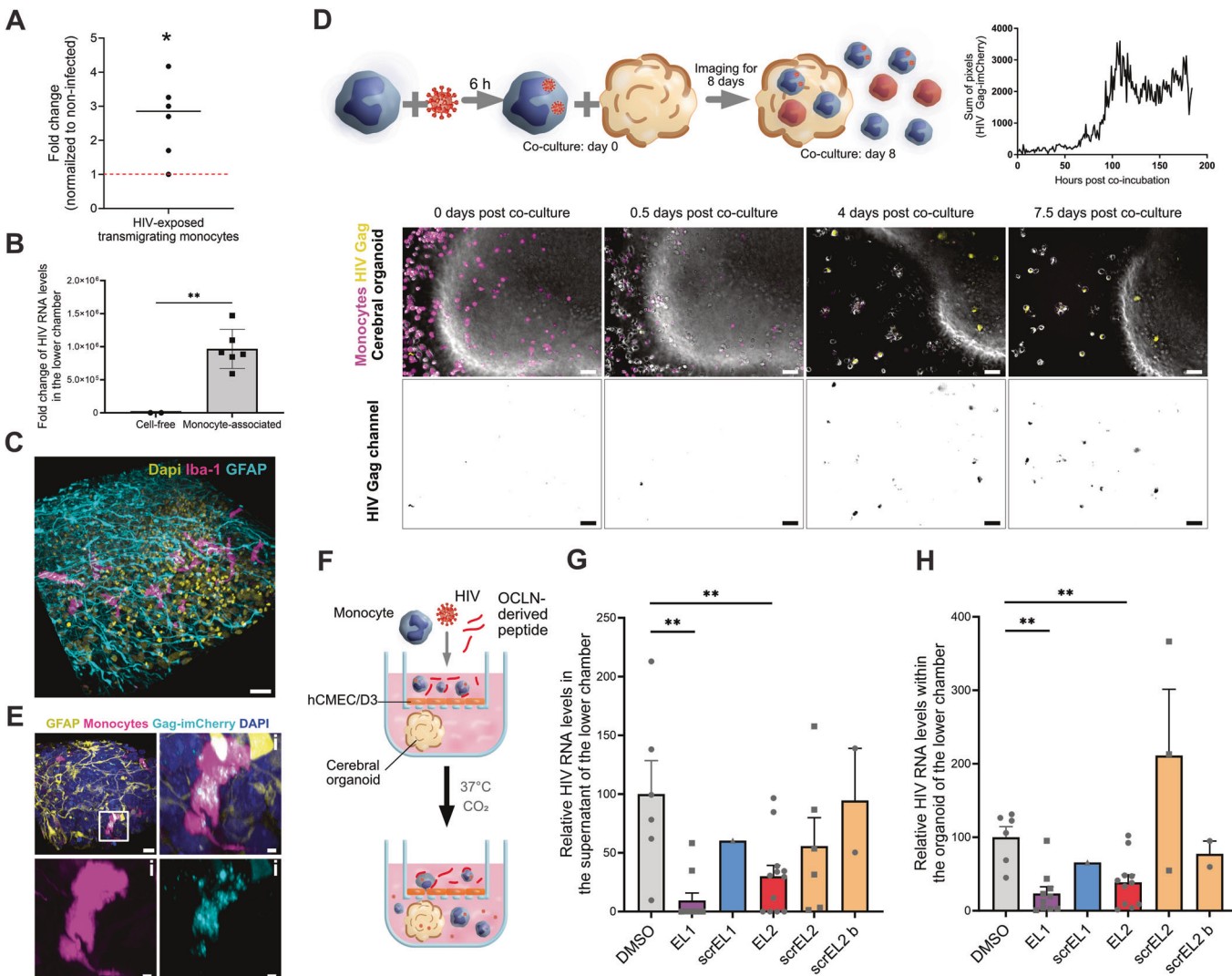

**Figure 4. OCLN-derived EL2 peptide inhibits HIV-1 neuroinvasion.**

(A) Primary monocytes were infected for 48 h at MOI 1 with HIV-1 (NLAD8) and added to the top chamber of transwells containing a hCMEC/D3 monolayer to allow transmigration to occur. After overnight incubation, the bottom chamber was harvested and the number of transmigrated monocytes was counted. The dot plot shows the mean of the fold change of the number of HIV-exposed monocytes over non-infected counterparts. Each dot corresponds to a donor. (B) Primary monocytes were infected for 24 h at MOI 1 with HIV-1 (NLAD8 (R5)). They were added to the top chamber of transwells containing a hCMEC/D3 monolayer to allow transmigration to occur at the same time as cell-free HIV (NLAD8 (R5)) was added to the top chamber of transwells (at the same amount as it was added to monocytes) containing a hCMEC/D3 monolayer for infiltration to happen. After overnight incubation, RNA from the top chamber and the bottom chamber was extracted. The data represents relative HIV RNA level in the bottom chamber (normalized to the top chamber) obtained from individual measurements. The data were obtained from $n = 2$ donors performed in three technical replicates. (C) Schematic of the experimental setup and image representation of time-lapse imaging of primary human monocytes pre-stained with CellTrace (magenta) and exposed to HIV-1 Gag-imCherry (yellow) for 6 h prior to the beginning of the co-incubation of monocytes with a hESC-derived cortical organoid. For representation purposes HIV-1 Gag-imCherry is shown separately in gray scale in the second row. Images were taken every hour for 8 days. At day 0 of co-incubation, monocytes carrying HIV-1 particles show yellow dots (not visible at this resolution). At 18 hpi (corresponding to 0.5 days post-co-incubation, second panel), very few cells were productively infected. Starting from day 4 post co-incubation (panel 3) up until day 7 (panel 4), multiple cells were observed to be productively infected and penetrating inside the cerebral organoid. Scale bar: 50 µm. Full videos can be found in Movie EV5 and EV6. This observation is supported by the graph with a pixel sum from a field of view of HIV-1 Gag-imCherry channel showing that pixel sum increases drastically at around 4 days (90–100 h) post co-incubation. (D) 3D reconstruction of an immunofluorescent image of a brain organoid with non-infected primary monocytes. Monocyte-derived cells are stained with Iba1 (magenta), astrocytes with GFAP (cyan), and cell nuclei with DAPI (yellow). Scale bar: 30 µm. (E) Three-dimensional reconstruction of an immunofluorescence image of a hESC-derived cortical organoid infiltrated with primary monocytes (CellTrace, magenta), infected with HIV-1 Gag-imCherry particles (cyan) for 7 days after the beginning of co-incubation (as in A). Astrocytes are stained with GFAP (yellow) and cell nuclei with DAPI (blue). Scale bar: 15 µm. The infected monocyte highlighted in the white square is magnified in (i). Scale bar: 5 µm. (F–H) Transmigration of primary monocytes exposed to HIV-1 R5 Gag-imCherry for 24 h, washed and pre-incubated with OCLN-derived peptides for 1 h. Monocytes were added at the top of a transwell with a confluent hCMEC/D3 monolayer and transmigration was allowed to happen overnight (≈16 h). (F) Schematic of the experimental setup. (G) Ratio of the amount of HIV RNA in the supernatant of the lower chamber was divided by the amount measured in the upper chamber. Each dot represents the ratio of the HIV RNA measurement performed in duplicate from a transwell containing a single organoid. The data represents were obtained from $n = 4$ individual experiments. Each dot represents a biological replicate (varies between conditions). Data normalized to DMSO of each experiment. (H) Ratio of the amount of HIV RNA in the organoid of the lower chamber was divided by the amount measured in the upper chamber. Each dot represents the ratio of the HIV RNA measurement performed in duplicate from a transwell containing a single organoid. Each dot represents a biological replicate (varies between conditions). Data normalized to DMSO of each experiment. Data information: In (A), the mean is represented with a black line and the normalization to non-infected (fold-change of 1) is marked by a red dotted line. In (B, G, H), the data are presented as mean ± SEM. Two-tailed Student's t-test $p$ value < 0.01 (**) and when not specified between conditions, the difference in not significant. Source data are available online for this figure.

Together, this data highlights that HIV-1 is carried by circulating monocytes into intracellular structures, while productive infection can be observed by the appearance of a diffuse cytosolic Gag staining.

## The OCLN-derived EL2 peptide prevents monocyte-mediated HIV neuroinvasion

To mimic monocyte-mediated HIV neuroinvasion, we took advantage of the monocyte-infiltrating cortical organoid assay we previously developed in the context of Zika virus neuroinfection (Ayala-Nunez et al, 2019). Briefly, we differentiated cerebral organoids from human embryonic stem cells (hESCs). We could observe that monocytes infiltrate cerebral organoids and exhibit Iba-1 positive labeling, a marker of microglial-like cells (Fig. 4C). Then, we incubated cerebral organoids with primary human monocytes previously exposed to fluorescent HIV-1 (Fig. 4D). Using long-term imaging, we were able to monitor in live the infiltration of primary monocytes carrying HIV-1 Gag-imCherry particles (Berre et al, 2013) for 8 days (Fig. 4D and Movie EV5 and Movie EV6). Here, we observed for the first time that monocytes carrying HIV-1 particles can infiltrate the organoid and started to show increased diffuse Gag-imCherry fluorescence (Fig. 4D, lower panels), suggesting that active viral replication had initiated. At high resolution in fixed samples, we detected infiltrated monocyte-derived cells productively infected by HIV-1 (puncta and cytosolic Gag-imCherry fluorescence) deep in the organoid (Fig. 4E). This data suggests that infiltration/differentiation of monocytes in the organoids is associated with the induction of productive HIV-1 replication.

Finally, we placed cortical organoids in the bottom chamber of hCMEC/D3-seeded transwells and added HIV-exposed primary monocytes treated with OCLN-derived peptides in the top chamber to allow transmigration (Fig. 4F). HIV RNA levels were measured in the supernatant of the bottom chamber as well as within organoids. In both conditions, we found that the treatment of monocytes with EL1 and EL2 peptides (in the upper chamber) significantly decreased the amount of HIV-1 RNA in both the supernatant of the lower chamber and within the organoid (Fig. 4G,H). In this context, the scramble peptide did not significantly modulate HIV RNA levels. Of note, relatively high heterogeneity was observed, which may be explained by the fact that each organoid grows in its own way, and although we used sized-matched organoids as much as possible, we cannot fully control the specific organoid composition in each condition. While EL1 did not show a significant inhibition of monocyte transmigration in our previous experiments performed in vitro and in vivo, it appears that in these experimental settings, the EL1 peptide strongly inhibited HIV neuroinvasion, suggesting that the OCLN-derived peptides may act on various steps to inhibit HIV neuroinvasion. For instance, one could envision that HIV propagation in cerebral organoids necessitates host factors that the EL1 peptide modulates, leading to an overall decrease of HIV RNA levels that is independent of HIV transmigration. OCLN was shown to directly regulate HIV transcription as well as nutrients needed for efficient HIV replication (Castro et al, 2016; Castro et al, 2018). Hence, one can hypothesize that EL1 and/or EL2 would exhibit antiviral activity against HIV, although this remains speculative at this stage. A team previously found that HIV-1 infection of monocytes is associated with increased expression of the TJAP molecules JAM-A and ALCAM (Veenstra et al, 2017). We were not able to see an increase of OCLN expression upon HIV-1 infection, but because hmOCLN acts as a general transmigration factor, rather than a virus-specific one, one could anticipate that targeting hmOCLN would also have an impact on HIV-exposed monocyte transmigration.

Together, our data indicates that the use of OCLN-derived peptides may represent a promising approach to decrease HIV neuroinvasion, and could help to decrease the burden of the CNS viral reservoir in seropositive patients. Here, we envision a strategy complementary to ART treatment, not aiming at targeting the virus, but rather at preventing it from reaching or replenishing its brain reservoir.

# Methods

**Reagents and tools table**

| Reagent/Resource | Reference or source | Identifier or catalog number |
|---|---|---|
| **Experimental Models** | | |
| THP-1 | ATCC | TIB-202 |
| hCMEC/D3 | Provided by Dr. Sandrine Bourdoulous | N/A |
| HEK 293T | ATCC | CRL-3216 |
| MEF | ThermoFisher Scientific | A34958 |
| Human PBMCs | Provided by Établissement Français du Sang | N/A |
| Primary human monocytes | Isolated from human PBMCs | N/A |
| GHOST X4/R5 cells | NIH AIDS reagents | 3942 |
| hESCs H9, WA09 | WiCell | WAe009-A |
| Tg(fli1a:eGFP-CAAX) zebrafish (*Danio rerio*) embryos | (Gebala et al, 2016) | ZDB-ALT-180504-1 |
| **Recombinant DNA** | | |
| pLenti-gRNA-OCLNx2-spCas9-T2A-Crimson-P2A-puro | This study; Addgene | 208398 |
| pLenti-gRNA-OCLNx2-spCas9-T2A-mNeonGreen-P2A-puro Addgene | This study; Addgene | 208400 |
| pLenti-gRNA-OCLNx2-spCas9-T2A-iRFP670-P2A-puro Addgene | This study; Addgene | 208399 |
| pTrip EGFP-OCLN | Addgene | 32789 |
| pTrip EGFP-OCLN-ΔC | Addgene | 32794 |
| pTrip EGFP-CAAX (Hras) | This study | N/A |
| pLenti-spCas9-T2A-Crimson-P2A-puro | Provided by Dr. Feng Zheng | N/A |
| psPAX2 (pSLQ8001, pCMV-R8.91) | Addgene | 202687 |
| pMD2.G (VSVG) | Addgene | 12259 |
| pNL(AD8) | NIH AIDS reagents | 11346 |
| pNL(AD8) HIV-1 Gag-iCherry | Berre et al, 2013 | N/A |
| **Antibodies** | | |
| Occludin Rb, pAb, WB, 1:1000 | GeneTex | GTX85016 |
| Occludin Rb, pAb, IF, 1:50 | Novus Biologicals | NBP1-87402 |

| Reagent/Resource | Reference or source | Identifier or catalog number |
|---|---|---|
| Rab11 Rb, mAb, IF, 1:50 | Cell Signaling | 5589 |
| Rab7 Rb, mAb, IF, 1:50 | Cell Signaling | 9367 |
| Rab13 Rb, pAb, IF, 1:50 | Novus Biologicals | NBP1-85799 |
| β-actin Rb, pAb, WB, 1:1000 | GeneTex | GTX109639 |
| GAPDH, WB, 1:1000 | GeneTex | GTX627408 |
| GFAP Ms, mAb, IF, 1:200 | Novus Biologicals | NB100-53809 |
| Iba1 Gt, pAb, IF, 1:200 | Novus Biologicals | NB100-1028 |
| VE-cadherin Dt, pAb, IF, 1:200 | R&D Systems | AF938 |
| HIV p17 Rb, pAb, IF, 1:100 | NIH AIDS reagents | 4811 |
| CD45-APC, FC, 1:50 | Miltenyi Biotec | 130-113-114 |
| EGFP Ms, mAb, WB, 1:1000 | ThermoFisher Scientific | MA5-15256 |
| anti-Mouse HRP, WB, 1:10,000 | ThermoFisher Scientific | SA1-100 |
| anti-Rabbit HRP, WB, 1:10,000 | ThermoFisher Scientific | 31458 |
| anti-Rabbit Alexa Fluor PLUS 488, IF, 1:1000 | ThermoFisher Scientific | A32790 |
| anti-Rabbit Alexa Fluor PLUS 555, IF, 1:1000 | ThermoFisher Scientific | A32794 |
| anti-Rabbit Alexa Fluor PLUS 647, IF, 1:1000 | ThermoFisher Scientific | A32795 |
| anti-Goat Alexa Fluor Plus 647, IF, 1:1000 | ThermoFisher Scientific | A32849 |
| anti-Mouse Alexa Fluor PLUS 555, IF, 1:1000 | ThermoFisher Scientific | A32773 |
| anti-Mouse Alexa Fluor PLUS 647, IF, 1:1000 | ThermoFisher Scientific | A32787 |
| **Oligonucleotides and sequence-based reagents** | | |
| siGENOME siRNA pool (RNAi Cherry-picked Library) | Dharmacon | LP_29911 |
| siGENOME Non-Targeting siRNA Pool #2 | Dharmacon | D-001206-15-05 |
| HIV gag F primer | Integrated DNA Technologies | See Methods |
| HIV gag R primer | Integrated DNA Technologies | See Methods |
| β-actin F primer | Integrated DNA Technologies | See Methods |
| β-actin R primer | Integrated DNA Technologies | See Methods |
| **Chemicals, Enzymes, and other reagents** | | |
| RPMI 1640 | Gibco®, ThermoFisher | 11875093 |
| DMEM | Gibco®, ThermoFisher | 11966025 |
| PBS | Gibco®, ThermoFisher | 10010023 |
| Neurobasal medium | Gibco®, ThermoFisher | 21103049 |
| N2 supplement | Gibco®, ThermoFisher | 17502048 |
| mTeSR Plus | STEMCELL Technologies | 05825 |
| Matrigel | Corning | 354277 |
| DMEM/F-12, GlutaMAX™ Supplement | Gibco®, ThermoFisher | 31331028 |
| KnockOut Serum Replacement | Gibco®, ThermoFisher | 10828028 |

| Reagent/Resource | Reference or source | Identifier or catalog number |
|---|---|---|
| GlutaMAX™ Supplement | Gibco®, ThermoFisher | 35050038 |
| MEM Non-Essential Amino Acid Solution | Gibco®, ThermoFisher | 11140050 |
| β-mercaptoethanol | Gibco®, ThermoFisher | 31350010 |
| IWR-1-Endo | Sigma-Aldrich | I0161 |
| Dorsomorphin | Sigma-Aldrich | P5499 |
| SB-431542 | Sigma-Aldrich | S4317 |
| Cyclopamine | Sigma-Aldrich | C4116 |
| EndoGro-MV | Millipore | SCME004 |
| Fetal bovine serum (FBS) | Sigma-Aldrich | F9665 |
| Penicillin-Streptomycin | Gibco®, ThermoFisher | 15140122 |
| b-FGF | Sigma-Aldrich | F0291 |
| Collagen, Type I solution from rat tail | Sigma-Aldrich | C3867 |
| Resveratrol | Sigma-Aldrich | R5010 |
| Lithium Chloride | Sigma-Aldrich | 62476 |
| Sodium Pyruvate | Sigma-Aldrich | S8636 |
| rhCCL2/MCP-1 protein | R&D Systems | 279-MC |
| BsmBI | ThermoFisher Scientific | ER0451 |
| Rapid DNA Ligation Kit | ThermoFisher Scientific | K1422 |
| FastAP | ThermoFisher Scientific | EF0654 |
| CalPhos™ Mammalian Transfection Kit | Takara Bio | 631312 |
| Sucrose | Sigma-Aldrich | S9378 |
| OCLN EL1-derived peptide | JPT Peptide Technologies or ProteoGenix, Tavelin et al, 2003 | See Methods |
| OCLN scrEL1 peptide | JPT Peptide Technologies or ProteoGenix | See Methods |
| OCLN EL2-derived peptide | JPT Peptide Technologies or ProteoGenix | See Methods |
| OCLN scrEL2 peptide a | JPT Peptide Technologies or ProteoGenix | See Methods |
| OCLN scrEL2 peptide b | ProteoGenix | See Methods |
| CellTiter-Glo Luminescent Cell Viability Assay | Promega | G7571 |
| Fibronectin | R&D Systems | 1918-FN |
| Triton X-100 | Sigma-Aldrich | X100 |
| Albumin Bovine BSA | Euromedex | 04-100-812-C |
| Human FcR blocking reagent | Miltenyi Biotec | 130-059-901 |
| RapiClear 1.52 reagent | Sunjin Lab | RC152001 |
| 3 kDa Dextran (Cascade Blue™) | ThermoFisher Scientific | D7132 |
| Protease inhibitor cocktail | Promega | G6521 |
| NuPage LDS Sample buffer | Invitrogen | NP0007 |
| Clarity ECL Substrate | Bio-Rad | 1705061 |
| Clarity Max ECL Substrate | Bio-Rad | 1705062 |
| Dithiothreitol (DTT) | ThermoFisher Scientific | 20290 |

| Reagent/Resource | Reference or source | Identifier or catalog number |
|---|---|---|
| MES Running Buffer | ThermoFisher Scientific | NP0002 |
| Tween 20 | Sigma-Aldrich | P1379 |
| Tris-HCl (Trizma® hydrochloride) | Sigma-Aldrich | T3253 |
| NaCl | Honeywell | S9888 |
| NP40 | Sigma-Aldrich | 492016 |
| Sodium deoxycholate | Sigma-Aldrich | D6750 |
| Sodium dodecyl sulfate (SDS) | Sigma-Aldrich | L3771 |
| Milk | Regilait | N/A |
| RNeasy Kit | Qiagen | 74106 |
| NucleoSpin RNA virus kit | Macherey-Nagel | 740956.50 |
| Luna Universal One-Step RT-qPCR Kit | New England Biolabs | E3005L |
| Lucifer yellow | Sigma-Aldrich | L0259 |
| CellTrace Calcein Red-Orange | Invitrogen | C34851 |
| CellTrace CFSE | Invitrogen | C34554 |
| LPS | Sigma-Aldrich | L3755 |
| **Software** | | |
| FUSION | Oxford instruments | N/A |
| Imaris software v9.7 | Bitplane, Oxford Instruments | N/A |
| Fiji upgrade of ImageJ | fiji.sc | N/A |
| FlowJo v10.8.1 | FlowJo LLC | N/A |
| GraphPad Prism 9 | GraphPad Software Inc. | N/A |
| SerialCloner 2-6-1 | https://serial-cloner.en.softonic.com/ | N/A |
| **Other** | | |
| ECM 830 Square Wave Electroporation System | BTX | 45-0052 |
| HTS Transwell®-96 Permeable Support 5.0 µm PC | Corning | 3387 |
| 24-Well Insert 5.0 µm PET clear | cellQUART®, Sabeu | 9325012 |
| Black 96-well µ-plate with a glass bottom | Ibidi | 89627 |
| 35 mm imaging dish with 4 compartments with a polymer coverslip bottom | Ibidi | 80416 |
| White cell-culture treated 96-well plate with a flat bottom | Corning | 3917 |
| Syringe filter, 0.45 µm pores | ClearLine | 146561 |
| Tissue culture orbital shaker (BENCHWAVER™ 3D rocker) | Benchmark Scientific | B3D5000 |
| 32 mL open-top thickwall polycarbonate ultracentrifuge tube | Beckman Coulter | 362305 |
| 38.5 ml thin-wall ultra-clear ultracentrifuge tube | Beckman Coulter | 344058 |
| SW32 Ti rotor | Beckman Coulter | 369694 |
| Optima XE-90 ultracentrifuge | Beckman Coulter | A94471 |

| Reagent/Resource | Reference or source | Identifier or catalog number |
|---|---|---|
| Andor Dragonfly Spinning Disk Microscope | Oxford Instruments | N/A |
| NovoCyte Flow Cytometry System | ACEA Biosciences | N/A |
| Waterbath ultrasonic cleaner | Velleman | 355666 |
| Trans-Blot Turbo Transfer System | Bio-Rad | LGOQBW15 |
| Bolt 4–12% Bis-Tris Plus Gels | ThermoFisher Scientific | NW04120BOX |
| Trans-Blot Turbo Mini PVDF membrane 0.2 µm | Bio-Rad | 1704156 |
| ChemiDoc Imaging System | Bio-Rad | OI91XQ15 |
| LightCycler 96 System | Roche | N/A |
| Infinite 200 Pro M Plex | Tecan | N/A |
| Nanoject microinjector 2 | Drummond | N/A |
| M205 FA stereomicroscope | Leica | N/A |
| Olympus Spinning Disk | Olympus Life Science | N/A |

## DNA constructs

An "all-in-one" Lentivector coding for two gRNAs targeting the OCLN gene, spCas9, E2-Crimson far-red fluorescent protein, and puromycin resistance was used. The initial plasmid backbone used was the pLenti-spCas9-T2A-Crimson-P2A-puro (kindly provided by Dr. Feng Zheng) and derived versions of this plasmid were also generated to express mNeonGreen (Addgene #122183), TagRFP-T (Addgene #122200) or iRFP670 (Addgene # 122182) fluorescent proteins. Here, the pLenti-spCas9-T2A-Crimson-P2A-puro was cleaved using the BsmBI restriction enzyme (Thermofisher) and treated with FastAP (Thermofisher). A synthetic DNA coding for BsmB1 enzyme, a first gRNA targeting the OCLN gene (5′-GGCCTCTTGAAAGTCCACCT-3′) under the control of the human U6 promoter, a tracrRNA for Cas9 recognition, a termination sequence, a second gRNA targeting the OCLN gene (5′-TGTCATC-CAGGCCTCTTGAA-3′) under the control of the hH1 promoter, a tracrRNA, and a BsmBI sequence was synthesized (Integrated DNA Technologies). The fragment was amplified using Phusion polymerase (Thermofisher). The insert was cloned into the prepared backbone using the T4 DNA fast ligase kit (Thermofisher) and the product was transformed in Stbl3 bacteria. The final lentivector pLenti-gRNA-OCLNx2-spCas9-T2A-Crimson-P2A-puro used in this study and depicted in Fig. EV1A is available on Addgene (#208398) as well as the variants pLenti-gRNA-OCLNx2-spCas9-T2A-mNeon-Green-P2A-puro Addgene (#208400) and pLenti-gRNA-OCLNx2-spCas9-T2A-iRFP670-P2A-puro Addgene (#208399). The pTrip EGFP-OCLN and EGFP-OCLN-ΔC lentiviral plasmids were obtained from Addgene (#32789 and #32794, respectively). The pTrip EGFP-CAAX plasmid was generated by amplifying the pTrip backbone from EGFP-OCLN including EGFP and the linker on one hand, and amplification of the CAAX sequence from mCherry-CAAX (Hras) (Addgene #108886) on the other hand. The fragments were assembled using Gibson Assembly (New England Biolabs).

## Cell culture

Human primary monocytes and THP-1 cell line (ATCC, TIB-202) were cultured in RPMI 1640 (Gibco) supplemented with 10% FBS (Sigma-Aldrich) and 1% Penicillin-Streptomycin (PS; Gibco). Human cerebellar microvascular endothelial cells D3 (hCMEC/D3) were kindly provided by Dr. Sandrine Bourdoulous (Institut Cochin, Paris, France). They were grown in EndoGro-MV Complete culture medium (Millipore) supplemented with 1 ng/mL b-FGF (Sigma-Aldrich) and 1% Penicillin-Streptomycin (Gibco) on 200 μg/mL rat collagen type 1-coated (Sigma-Aldrich) plates. For experimental use of hCMEC/D3 cells, EndoGro-MV complete medium was additionally supplemented with 10 μM Resveratrol (Sigma) and 10 mM Lithium Chloride (Merck Millipore); cells were differentiated for 7 days prior to an experiment. HEK 293T cells (ATCC, CRL-3216) were cultured in DMEM (Gibco) supplemented with 10% FBS and 1% PS. Mouse fibroblast cells (MEF, Thermo Fisher Scientific, A34958) feeder cells were kept in DMEM with 10% FBS, 1% PS, and 1 mM Sodium Pyruvate (Sigma). All cells were kept at 37 °C in 5% $CO_2$. All cell lines were frequently tested for mycoplasma contamination, but were not recently authenticated.

## Human monocyte isolation

Peripheral blood mononuclear cells (PBMCs) were isolated from buffy coats of healthy blood donors provided by Établissement Français du Sang (EFS; agreement #21PLER2019-0106). All donors signed a consent form allowing the use of their blood for research purposes. PBMCs were purified on a density gradient with Ficoll-Paque (Cytiva). Monocytes were isolated with CD14+ microbeads and LS columns (Miltenyi Biotec). Monocytes from a total of 32 donors were used.

## Transcriptomic analysis of monocytic TJAPs

RNA-seq data from primary human monocytes and THP-1 cells were downloaded from the Gene Expression Omnibus (GEO) (Edgar et al, 2002). RNA-seq read counts from four monocyte donors (accession ID: GSE74246) were further normalized to reads per kilobase of exon model per million mapped reads (RPKM) in R. The RNA-seq data from THP-1 cells (accession ID: GSM927668) was already available in RPKM. The heatmap was created in Excel.

## SiRNA-based THP-1 transmigration screen

Monocytic TJAPs were knocked down (KD) in THP-1 cells via electroporation using a siGENOME siRNA pool (RNAi Cherry-picked Library, Dharmacon LP_29911). A non-targeting siRNA (siGENOME Non-Targeting siRNA Pool #2, Dharmacon D-001206-15-05) was used as a control (siCtrl). THP-1 cells were counted, centrifuged (1000 rpm, 5 min, RT), and suspended in cold PBS to a cell density of $5 \times 10^6$ cells/ml. The cells were then aliquoted in Eppendorf tubes at a concentration of $10^6$ cells/tube in 200 μL. The siRNA was added to a final concentration of 200 nM. The cells were gently mixed with the siRNA and incubated for 5 min on ice. The cell/siRNA mixture was transferred to an electroporation cuvette (2 mm). The cells were electroporated at RT with the following parameters: 600 V, 50 μs, 4 pulses, interval 1 s using an ECM 830 Square Wave Electroporation System (BTX No. 45-0052). The electroporated cells were taken out of the cuvette

using a sterile plastic Pasteur pipette, placed in Eppendof tubes, and incubated for 10 min on ice. Finally, the cells were seeded in 12-wells plates containing cell culture media without antibiotics (1.5 ml/well of RPMI 10% FBS, 1 mM Sodium pyruvate). Non-electroporated cells were used as negative control.

The transmigration experiment was performed in 96-wells transwells (Corning, 5 μm pore size, PC). The hCMEC/D3 cells were grown as described above. The KD THP-1 cells were harvested after 3 days of electroporation. A total of 150,000 cells were added per transwell in a volume of 100 μL of cell culture media (RPMI, 10% FBS, 1 mM sodium pyruvate, 1% PS), while freshly prepared media (RPMI, 10% FBS, 1% PS) supplemented with 200 ng/ml MCP-1 (R&D Systems) was added to the bottom chamber. After an overnight incubation (~16 h), the transmigrated cells present in the bottom chamber were harvested and counted. The percentage of transmigrated cells was calculated with respect to the siCtrl condition.

## Lentivirus production

All viruses and lentiviruses were produced by transfecting HEK 293T cells using CalPhos Mammalian Transfection Kit (Takara Bio) according to the manual, in DMEM, 10% FBS, 1% PS. For viral production, a single plasmid was used (100–200 ng/cm²). For lentiviral production, the ratio of 2:1:2 was used for construct-of-interest: VSVG:psPAX2 or VSVG:pSIV3 + 1:2 (200–300 ng/cm² total). Overnight after transfection, medium was changed to fresh DMEM, 10% FBS, 1% PS. Medium containing viruses or lentiviruses were harvested 48 h later. It was spun down ($1000 \times g$, 5 min) and filtered through a syringe filter with 0.45 μm pores. Then, the supernatant was transferred onto 10% sucrose (Sigma) in PBS inside a 32 mL open-top thickwall polycarbonate ultracentrifuge tube (Beckman Coulter) and centrifuged using SW32 Ti rotor in Optima XE-90 ultracentrifuge (Beckman Coulter) at 28,000 rpm for 2 h at +4 °C. The supernatant was discarded and pellets were resuspended in cold RPMI (1/100) on ice. After 1 h of incubation on ice, RPMI with viruses/lentiviruses was frozen at −80 °C until used. Further transduction of monocytes was done by co-transduction of a lentivirus containing a construct of interest and a lentivirus with a Vpx-encoding construct (pSIV3+). The titer of lentiviruses was assessed by flow cytometry using the fluorescent tag of the protein of interest.

## HIV production and titration

Plasmids NLAD8 HIV-1 AD8 Macrophage-Tropic R5 (NIH AIDS reagents #11346) or HIV Gag-imCherry V3 loop (Berre et al, 2013) were transfected in HEK 293T cells using CalPhos Mammalian Transfection Kit (Takara Bio) in DMEM, 10% FBS, 1% PS, according to the manufacturer's instructions. After overnight incubation, the media was changed to fresh media. The supernatant containing the viruses was harvested 72 h later. It was spun down ($1000 \times g$, 5 min) and filtered through a syringe filter with 0.45 μm pores. Then, supernatant was transferred onto 10% sucrose (Sigma-Aldrich) in PBS in a 38.5 ml thin-wall ultra-clear ultracentrifuge tube (Beckman Coulter) and ultracentrifuged using SW32 Ti rotor in Optima XE-90 ultracentrifuge (Beckman Coulter) at 28,000 rpm for 2 h at 4 °C. Supernatant was discarded and pellets were resuspended in cold RPMI on ice for 4 h. Virus stocks were frozen

at −80 °C until used. The titer of viruses was assessed at 48 hpi by flow cytometry on GHOST X4/R5 cells (NIH AIDS Reagents # 3942; (Morner et al, 1999)).

## Biosafety

Experiments performed with cells or organoids were performed in biosafety laboratory level 2 (BSL-2) and experiments performed with HIV-1 were performed in BSL-3.

## Peptides

OCLN-derived peptides were designed to represent a part of each extracellular loop (EL) of OCLN. EL1-derived peptide: DRGYGTSLLGGSVGYPYGGSGFGS and EL2-derived peptide: YGSQIYALCNQFYTPAATGLYVD. They were generated based on a study from Tavelin et al (Tavelin et al, 2003). They were designed to cover the potential site of Occludin-Occludin interaction. Scramble peptides were generated by randomly reordering the amino acids of the peptides. The scrEL1 used in this study was: GGTSPLYGGFVGGYSGDSYGRGSL, the scrEL2 a peptide was: PQAYDFTGNGSYLCTLYAYVIAQ, and the scrEL2 b peptide was LIFYATQCYDATYVNGPLAQYSG. The peptides were synthesized by JPT Peptide Technologies and ProteoGenix.

### Transmigration assay

For transmigration experiments which do not involve HIV, hCMEC/D3 cells were seeded in cellQART 24-well transwell inserts (Sabeu, 5 µm pore size PET) at 30,000 cells/insert and grown for 7 days on collagen as described above. Primary monocytes or THP-1 cells were added at the top of the transwell (200,000 cells/well, 0.2 ml of RPMI, 10% FBS, 1% PS) while freshly prepared media (RPMI, 10%, FBS 1% PS) supplemented with 200 ng/ml MCP-1 (R&D Systems) was added to the bottom chamber. Monocytes or THP-1 cells were transduced 48 h before the experiment, washed in RPMI 10% FBS, 1% PS once after overnight incubation and once before the transmigration experiment. In the case of OCLN-derived peptide, monocytes were washed in RPMI 10% FBS, 1% PS once after overnight incubation, incubated with OCLN-derived peptides and the monocyte-peptide mixture was used the next day for the transmigration experiment. The transmigration through hCMEC/D3 cells was performed overnight, and cells from the bottom and the top chambers were recovered and prepared for flow cytometry analysis.

For transmigration experiments which involve HIV, hCMEC/D3 cells were seeded in 96-well transwells (Corning, 5 µm pore size, PC) at 15,000 cells/insert and grown for 7 days on collagen as previously described. Monocytes were exposed to HIV-1 for 24 h and washed at least twice before the experiment to reduce the possibility of cell-free viral infiltration. After 7 days, hCMEC/D3 cells were washed, and previously HIV-exposed monocytes were added at the top of the transwell after 1 h incubation with OCLN-derived peptides (100,000 cells/well, 0.1 mL of RPMI, 10% FBS, 1% PS). The bottom chamber was filled either with 0.22 ml of fresh media (RPMI, 10% FBS, 1% PS) supplemented with 200 ng/mL of MCP-1 or with a 35+ days-old brain organoid in 0.22 ml of Neurobasal medium with N2 supplement, 1x GlutaMAX, 1% PS. After 24 h of incubation the top and bottom chambers were lysed for RNA extraction and RT-qPCR. For the top chamber all the cells

(monocytes and endothelia) were lysed and analyzed together. For the bottom chamber, supernatant and organoids were separated after monocyte transmigration for further analysis. Supernatants together with any monocytes were lysed and analyzed together (this procedure does not allow for the discrimination of cell-free monocyte-associated viral RNA). Organoids were washed 3 times with PBS, lysed, and viral RNA analyzed by RT-qPCR.

## Adhesion assay

The readout for the adhesion assay was measured either with the CellTiter-Glo Luminescent Cell Viability Assay (Promega) or by flow cytometry. For luminescence, readout was performed in a white cell-culture treated 96-well plate with a flat bottom (Corning). The coating of the bottom of the plate was done with 10 µg/mL of human fibronectin (R&D Systems). Monocytes were added to the coated substrate and incubated for 30 min at 37 °C. Then, monocytes were washed 2× with PBS to remove non-adherent cells. The cells that remained attached were analyzed with the CellTiter-Glo. For flow cytometry, Monocytes were added to flat bottom clear cell-culture treated 96-well plates coated with 10 µg/mL human fibronectin or hCMEC/D3 cells and incubated for 30 min at 37 °C. Then, both suspension cells and adherent cells were stained for CD45+ (or not stained if they expressed EGFP) and analyzed by flow cytometry.

## Immunofluorescence and image acquisition

THP-1 cells or primary human monocytes were seeded onto 12 mm glass coverslips in 24-well plates, with either a 10 µg/ml fibronectin coating or a confluent monolayer of adherent hCMEC/D3 cells, and fixed in 4% PFA for 15 min at room temperature (RT). Cells were permeabilized with 0.05% triton X-100 (Sigma-Aldrich) with 0.5% BSA (Euromedex) for 15 min, then blocked in 10% FBS (Sigma-Aldrich) with 1:5 human FcR blocking reagent (Miltenyi Biotec) for 20 min. Primary antibody staining was performed for 1 h in PBS (Gibco), followed by washes and secondary antibody staining for 1 h. Images were acquired using an Andor Dragonfly Spinning Disk Microscope (Oxford Instruments) equipped with a 1024 × 1024 EMCCD camera (iXon Life 888, Andor) with either a ×40, ×63, or ×100 objectives. Pre-processing of images was performed in FUSION (Oxford Instruments).

The hESC-derived cerebral organoids were fixed in 4% PFA overnight at 4 °C and permeabilized with 0.5% triton X-100 with 0.5% BSA for 24 h. Blocking was done with 0.5% BSA supplemented with 1:5 human FcR blocking reagent for 4 h followed by 24 h staining firstly with primary antibodies and then secondaries in 0.5% triton X-100 with 0.5% BSA. Organoids were transferred to a black 96-well µ-plate (Ibidi) and cleared in RapiClear 1.52 reagent (Sunjin Lab) to perform deep fluorescence imaging. Images were acquired using an Andor Dragonfly Spinning Disk Microscope.

## Live-cell imaging

For transmigration experiments, hCMEC/D3 cells were grown for 7 days in their experimental medium on collagen as described above in a 35 mm imaging dish with 4 compartments with a polymer coverslip bottom (Ibidi), then stained with CellTrace Yellow (Invitrogen) 1 h before the beginning of imaging for 20 min and

washed 3 times with RPMI 10% FBS, 1% PS. THP-1 cells or primary human monocytes previously transduced with GFP+-constructs were put on pre-stained hCMEC/D3 cells 30 min before the beginning of imaging. For fluid phase marker imaging, fluorescent Dextran was used as previously (Gaudin et al, 2013). Briefly, monocytes were added onto hCMEC/D3 cells 1 h before imaging. At the beginning of imaging, 100 μg/ml of 3 kDa fluorescent Dextran was added to the cells. Imaging was done using Andor Dragonfly Spinning Disk Microscope as above with ×100 objective (frame rate for transmigration experiments: 10 min, z step: 0.8 μm; frame rate for fluid phase imaging: 1 min, z step: 0.5 μm).

## Image processing and quantification

Image processing was performed in Imaris software v9.7 (Bitplane, Oxford Instruments) or the Fiji upgrade of ImageJ. Co-localization or number of pixels per field per time point were assessed using Imaris software using standard procedures. Briefly, for co-localization analysis, images were thresholded using EGFP fluorescence intensity to remove background and diffuse signal. Threshold was kept the same per condition and experiment repeats. Pearson correlation coefficient was and plotted for comparative analyses. For pixel anaysis, all images were thresholded the same way and data was extracted and plotted.

## Flow cytometry

Cells expressing fluorescent proteins were fixed in 4% PFA and washed in PBS before flow cytometry. When immunostaining was used, live samples were blocked with 10% FBS in PBS and 1:5 FcR Blocking Reagent. To label cell surface proteins, samples were stained with primary conjugated antibodies on ice for 1 h and washed 2× with PBS. Then, the cells were fixed in 4% PFA and washed in PBS before flow cytometry. Data were acquired with NovoCyte Flow Cytometry System (ACEA Biosciences). Data was analyzed using FlowJo (LLC).

## Western blotting

Cells were lysed in RIPA buffer at about $10^6$ cells in 50 μL (50 mM Tris-HCl (Sigma-Aldrich) pH 8.0, 150 mM NaCl (Honeywell), 1% NP40 (Sigma), 0.5% sodium deoxycholate (Sigma-Aldrich), 0.1% SDS (Sigma-Aldrich), freshly added protease inhibitor cocktail 1x (Promega)). Samples were sonicated at 4 °C in a waterbath ultrasonic cleaner (Velleman) for 8 min, denatured in NuPage LDS Sample buffer (Invitrogen) supplemented with DTT (Thermofisher) at final concentration 100 mM for 3 min at 98 °C. Samples were run on Bolt 4–12% Bis-Tris Plus Gels (Thermofisher) at 140 V for 1.5 h in MES Running Buffer (Thermofisher), then transferred onto Trans-Blot Turbo Mini PVDF membrane 0.2 μm (Bio-Rad) using Trans-Blot Turbo Transfer System (Bio-Rad). Samples were blocked in 5% milk (Regilait) in TBS supplemented with 0.05% Tween 20 (Sigma-Aldrich; TBST). Membranes were incubated with primary antibodies at 4 °C overnight in TBST 0.05% with 5% BSA (Euromedex), washed 3 × 10 min with TBST 0.05%, incubated with secondary HRP antibodies for 1 h at RT, washed 3 × 10 min with TBST 0.05% and revealed using either Clarity or Clarity Max ECL Substrate (Bio-Rad). Chemiluminescence imaging was performed using a ChemiDoc Imaging System (Bio-Rad).

## RT-qPCR

RNA was extracted either from cells and organoids using the RNeasy Kit (Qiagen) or from supernatants using NucleoSpin RNA virus kit (Macherey-Nagel). Reverse transcription and quantitative real-time PCR was performed using Luna Universal One-Step RT-qPCR Kit (New England Biolabs) in a 96-well plate and qPCR was performed in a LightCycler 96 System (Roche). Data was not normalized to β-actin because only HIV relative quantity was needed. β-actin was used to assess the presence of cells. HIV gag primers: 5′-ACTCTAAGAGCC-GAGCAAGCT-3′, 5′-TCTAGTGTCGCTCCTGGTCC-3′. β-actin primers: 5′-CACCATTGGCAATGAGCGGTTC-3′, 5′-AGGTCTTT GCGGATGTCCACGT-3′.

## Lucifer yellow permeability assay

hCMEC/D3 cells were cultured as described before for 7 days in transwells before any experimental manipulations. For monitoring hCMEC/D3 permeability with Occludin-derived peptides, peptides were added overnight and not washed away before Lucifer yellow addition. Permeability was assessed by adding 50 μM Lucifer yellow in the top chamber. After 2 h incubation inside the incubator, media was collected from the top and bottom chambers of transwells. Quantification of fluorescence was done using a microplate reader (Infinite 200 Pro M Plex, Tecan) with excitation at 428 nm and emission at 536 nm.

## Zebrafish embryo permeability and transmigration assay

*Tg(fli1a:eGFP-CAAX)* zebrafish (*Danio rerio*) embryos were used in all experiments. All animal procedures were performed in accordance with French and European Union animal welfare guidelines and supervised by local ethics committee (Animal facility #A6748233; APAFIS #2018092515234191). Embryos were maintained in Danieau 0.3X medium (17.4 mM NaCl, 0.2 mM KCl, 0.1 mM $MgSO_4$, and 0.2 mM $Ca(NO_3)_2$) buffered with 0.15 mM HEPES (pH 7.6) and supplemented with 200 mM of 1-Phenyl-2-thiourea (Sigma-Aldrich) to inhibit the melanogenesis. For all zebrafish experiments, the offspring was selected based on anatomical/developmental good health. Embryos were split randomly between experimental groups. Forty-eight hours post fertilization (hpf), embryos were mounted in a 0.8% low-melting-point agarose pad containing 650 mM of tricain (ethyl-3-amino-benzoate-methanesulfonate) to immobilize them. Primary human monocytes frozen at $10 \times 10^6$ cells/mL were thawed 2 h and seeded at $5 \times 10^6$ cells/mL before their injection in zebrafish. Monocytes were labeled with CellTrace Calcein Red-Orange (Invitrogen) according to the manufacturer's instructions. Monocytes were resuspended in PBS in which the peptides EL1 or EL2 or DMSO were diluted at 1:100. Monocytes were injected with a Nanoject microinjector 2 (Drummond) and microforged glass capillaries (20-μm inner diameter) filled with mineral oil (Sigma-Aldrich), and 18.4 nL of a cell suspension at $10^8$ cells/mL were injected in the duct of Cuvier of the embryos under a M205 FA stereomicroscope (Leica). Injected embryos were maintained at 28 °C in between injection and imaging. At 6–8 h post injection, confocal imaging was performed with an inverted Olympus Spinning Disk (×30 objective, NA 1.05). Image analysis and processing were performed using ImageJ. Permeabilization control was performed using LPS

(Sigma-Aldrich) diluted at 50 µg/ml or 100 µg/ml in the breeding water (Danieau 0.3X + PTU) of the embryos after mounting and 2 h before injections. Dextran 70 kDa coupled to TRITC (Invitrogen) was injected in the duct of Cuvier and embryos were incubated at 28 °C for 15 min before imaging to permit dextran diffusion in the embryos. Images were taken with a M205 FA stereomicroscope equipped with a DFC3000G CCD camera (Leica Microsystems).

## Stem cell culture and cortical organoid differentiation

H9 human embryonic stem cells (hESCs; female, WA09, WiCell) were cultured in mTeSR Plus medium (STEMCELL Technologies) with 1% Penicillin-Streptomycin in 8 mg/mL Matrigel-coated (Corning) 6 cm dishes (Corning). The medium was changed daily and cells were split by cutting colonies every 5–6 days. For hESC-derived cerebral organoid differentiation hESCs were transferred while splitting on MEF feeder layer in ESC medium (DMEM F12 + L-glutamine (Gibco), 20% KnockOut Serum Replacement (Gibco), 1x MEM Non-Essential Amino Acid Solution (Gibco), 100 µM β-mercaptoethanol (Gibco), 1% PS) with freshly added 8 ng/mL human bFGF (Sigma-Aldrich). The medium was changed daily. When hESC colonies reached 1.5–2 mm, medium was replaced with differentiation medium (ESC medium with 1 mM Sodium Pyruvate, no bFGF), then they were picked and lifted using a cell lifter (Corning) and transferred to a 6 cm ultra-low attachment dish (Corning) together with differentiation media. The medium was replaced every other day. On day 0 (24 h post-lifting, when embryoid bodies were formed) half of the medium was replaced with differentiation medium supplemented with 3 µM IWR-1-Endo (Sigma), 1 µM Dorsomorphin (Sigma), 10 µM SB-431542 (Sigma), and 1 µM Cyclopamine (Sigma) final concentration. This supplemented medium was maintained and changed every other day. On day 3, differentiating organoids were placed on a tissue-culture orbital shaker in a tissue culture incubator. On day 18, medium was switched to Neurobasal medium (Gibco) with 1x N2 supplement (Gibco), 1x GlutaMAX (Gibco), 1% PS, and 1 µM Cyclopamine (Sigma), changed every other day. On day 24, Cyclopamine was removed from the content of the medium. Organoids were taken for experiments starting from day 35.

## Bioinformatic analysis of zebrafish RNAseq data

The RNAseq data from zebrafish endothelium come from Bonkhofer et al (Bonkhofer et al, 2019). Quality control was checked by FastQC v0.11.5 and reads were trimmed using trim_galore (version 0.4.4 options -q 20 –stringency 2) and mapped to the Zebrafish Genome Zv11 genome using Hisat2 v2.0.5 with Ensembl transcriptome index. Bigwig files were generated using bam2wig.py from the RSeQC package v2.4 (parameters -u -t 5000000000) and visualized in the Integrative Genomics Viewer (IGV). For gene expression analysis, unique reads were counted in Ensembl genes with HTSeq v0.9.1 (parameters –t exon –s no). The FPKM values were generated using DESeq2.

## Statistical analysis

Student's unpaired t test was performed as described in figure legends, except for zebrafish experiments where Kruskal–Wallis test with Dunn's post hoc test was done. Analyses were done using GraphPad Prism (v9.5.1).

## Data availability

No primary datasets have been generated.

The source data of this paper are collected in the following database record: biostudies:S-SCDT-10_1038-S44319-024-00190-x.

## Peer review information

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

## Acknowledgements

We acknowledge the MRI imaging facility, part of Biocampus Montpellier, member of the national infrastructure France-BioImaging, for advice and training. The following reagent was obtained through the NIH HIV Reagent Program, Division of AIDS, NIAID: pNLAD8 HIV-1 AD8 Macrophage-Tropic R5 (11346); GHOST CXCR4 + CCR5+ Cells (#ARP-3942), contributed by Dr. Vineet N. KewalRamani and Dr. Dan R. Littman. We thank Philippe Benaroch (Institut Curie, Paris, France) from providing the HIV Gag-imCherry V3 loop DNA plasmid. We thank Isis Ricaño-Ponce (Innova Bio Data) for her help with the RNAseq data analyses in Fig. 1A, and we thank Ambre Bender and Michael Dumas (CNRS, UMR7242, Illkirch) for their help with RNAseq data analysis in Fig. EV4A. Aude Boulay for helping with blood samples, and Lucie Klughertz for HIV-1-related experiments. This work was supported by the ANRS MIE to DB, NVAN, and RG, Sidaction to DB and RG, the Fondation pour la Recherche Médicale (FRM) to VL, the Agence Nationale de la Recherche (ANR; ANR-20-CE15-0019-01) to RG, the French embassy in Slovakia and the Program Hubert Curien (PHC) Stefanik to VL and RG. AD and VM are supported by University of Strasbourg and ARC (Association pour la Recherche contre le Cancer) to VM. Work in the lab of JGG (NO) is supported by INSERM and University of Strasbourg; the team receives recurrent support from La Ligue contre le Cancer.

## Author contributions

**Diana Brychka**: Resources; Formal analysis; Investigation; Visualization; Methodology; Writing—review and editing. **Nilda Vanesa Ayala-Nunez**: Formal analysis; Investigation; Methodology. **Amandine Dupas**: Formal analysis; Validation; Investigation; Visualization; Methodology. **Yonis Bare**: Formal analysis; Visualization; Methodology. **Emma Partiot**: Methodology. **Vincent Mittelheisser**: Resources; Methodology. **Vincent Lucansky**: Resources; Methodology. **Jacky G Goetz**: Supervision; Funding acquisition; Visualization; Methodology; Writing—review and editing. **Naël Osmani**: Formal analysis; Supervision; Investigation; Visualization; Methodology; Writing—review and editing. **Raphael Gaudin**: Conceptualization; Resources; Formal analysis; Supervision; Funding acquisition; Validation; Investigation; Visualization; Methodology; Writing—original draft; Project administration; Writing—review and editing.

Source data underlying figure panels in this paper may have individual authorship assigned. Where available, figure panel/source data authorship is listed in the following database record: biostudies:S-SCDT-10_1038-S44319-024-00190-x.

## Disclosure and competing interests statement

The authors declare no competing interests.

# Expanded View Figures

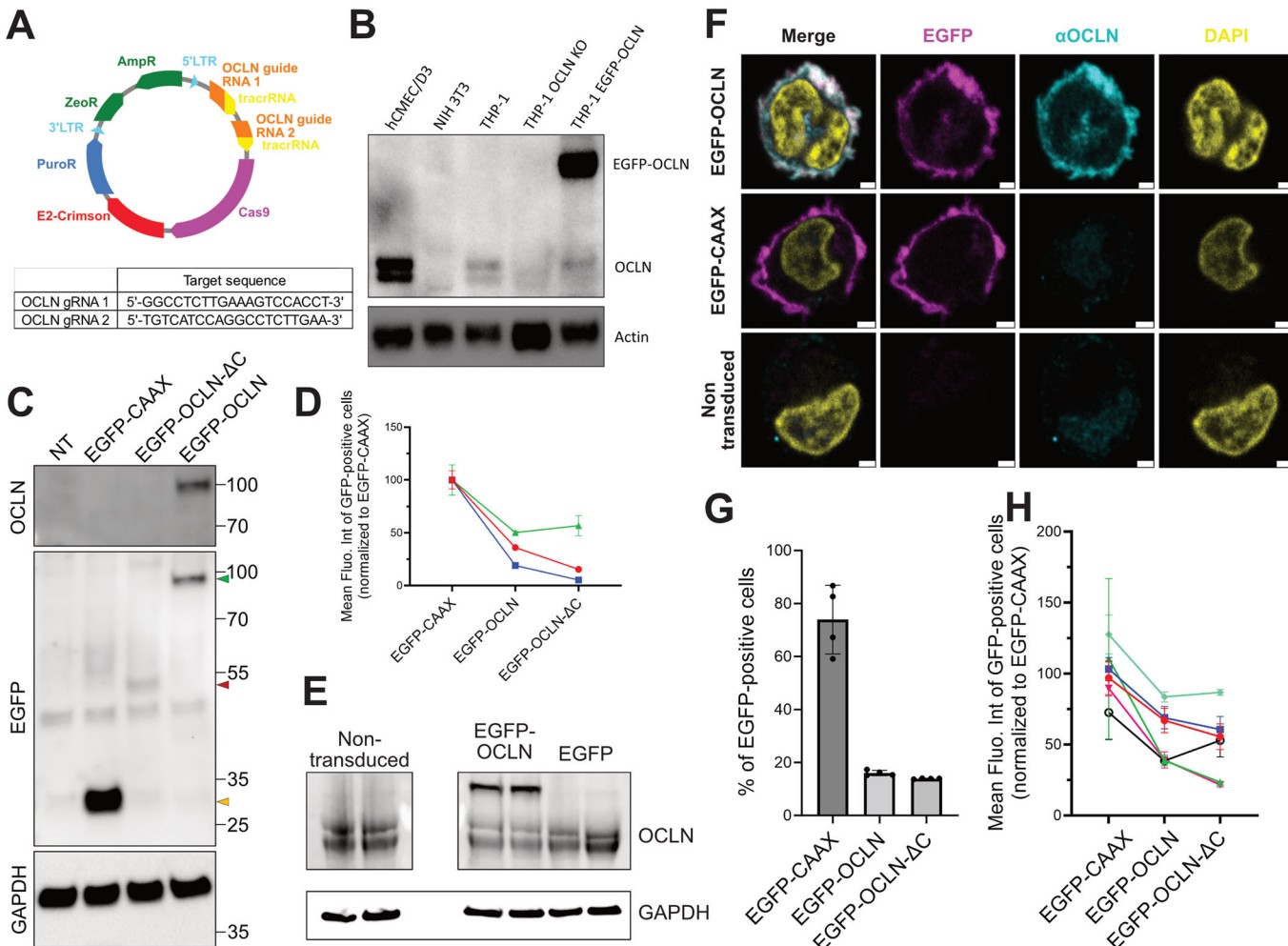

**Figure EV1. Characterization of monocytic Occludin expression.**

(A) Schematic representation of a CRISPR/Cas9 lentiviral construct used for the generation of OCLN KO THP-1 cells with indicated target sequences. (B) Western Blot analysis of OCLN expression in indicated cell lines. Actin is used as a loading control. The hCMEC/D3 and THP-1 cells express OCLN (two bands), the THP-1 OCLN KO cells do not show OCLN, although non-specific bands of lower size and very weak intensity can be observed. As controls, NIH 3T3 do not express OCLN and THP-1 WT transduced with EGFP-OCLN express both endogenous and overexpressed OCLN. (C) Western Blot analysis of OCLN expression in THP-1 OCLN KO cells. GAPDH is used as a loading control. EGFP-OCLN can be seen with both anti-Occludin and anti-GFP antibody (green arrow at EGFP). EGFP-OCLN-ΔC and EGFP-CAAX can be seen with anti-GFP antibody only (red and yellow arrows, respectively). (D) Mean fluorescence intensity of EGFP-CAAX, EGFP-OCLN, or EGFP-OCLN-ΔC expressed by THP-1 OCLN KO cells measured by flow cytometry. The data were normalized to EGFP-CAAX obtained from $n = 3$ individual experiments. Each symbol's color corresponds to an individual experiment. (E) Western Blot showing endogenous OCLN expression in human primary monocytes and exogenous expression of EGFP-OCLN in the same donors (2 donors). GAPDH is used as a loading control. (F) Immunofluorescence images of primary monocytes transduced with EGFP-OCLN, EGFP-CAAX, or non-transduced, fixed and stained with anti-OCLN antibody (cyan) and DAPI (yellow). EGFP is in magenta. Scale bar: 2 μm. (G) Efficiency of transduction of primary monocytes from 2 donors performed in two technical replicates measured by flow cytometry. (H) Mean fluorescence intensity of EGFP-CAAX, EGFP-OCLN or EGFP-OCLN-ΔC expressed by primary monocytes. The data were obtained from $n = 6$ monocyte donors normalized to EGFP-CAAX of each experiment. Each symbol's color corresponds to an individual monocyte donor. In (D, G, H), the data are presented as mean ± SEM. Source data are available online for this figure.

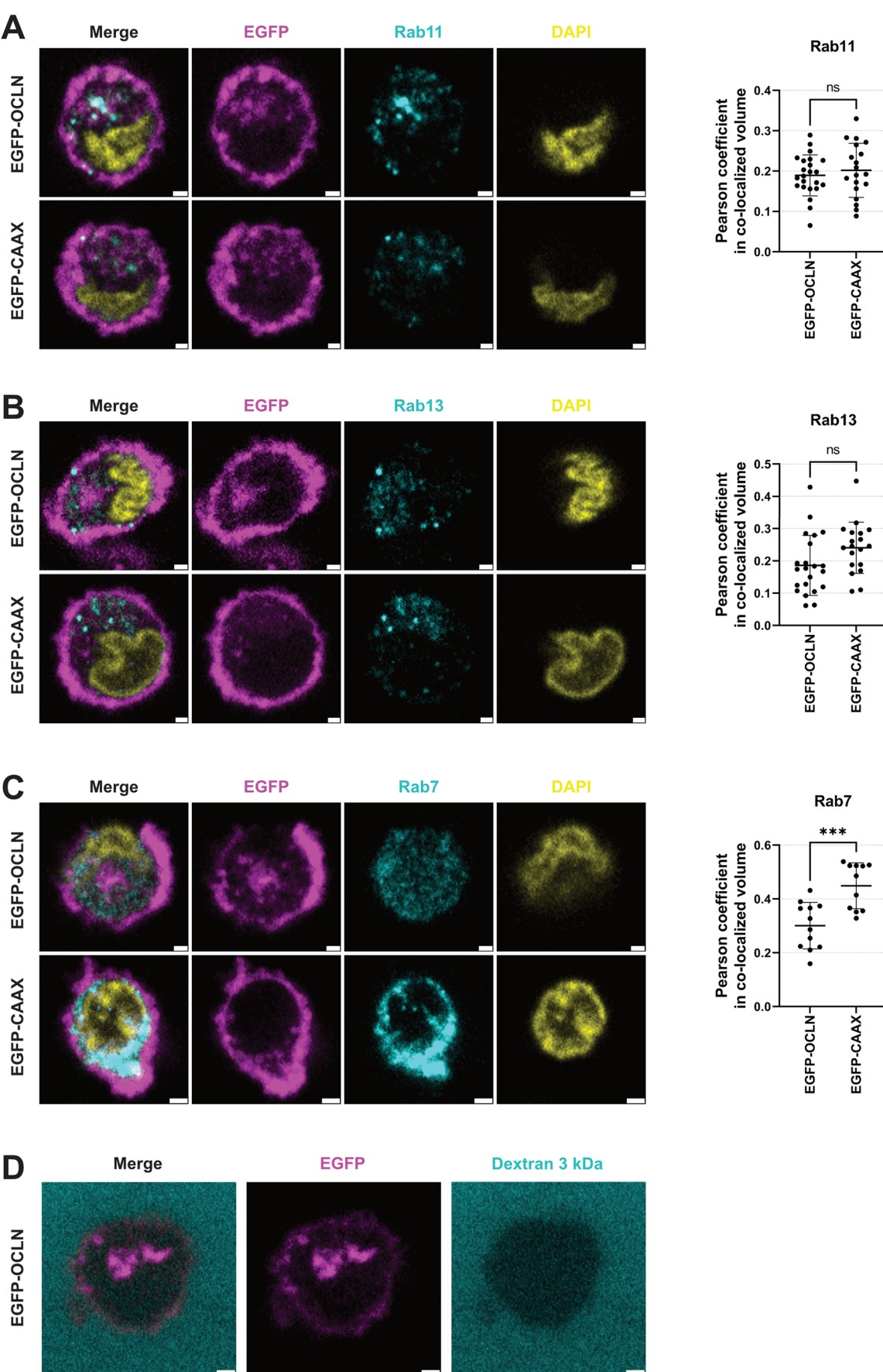

◀  **Figure EV2.  Characterization of the OCLN-containing compartment.**

(**A–C**) Immunofluorescence image of primary monocytes expressing either EGFP-OCLN or EGFP-CAAX (magenta) attached to hCMEC/D3 cells and stained with DAPI (yellow) and antibodies (cyan) against Rab11 (**A**), Rab13 (**B**), or Rab7 (**C**). Scale bar: 1 μm, except in C for EGFP-CAAX: 2 μm. (**D**) Primary monocyte expressing EGFP-OCLN (magenta) attached to hCMEC/D3 monolayer were incubated with 3 kDa fluorescent Dextran (cyan) and immediately imaged using confocal microscopy. The snapshots highlight that Dextran does not access the OCLN-containing compartment. Scale bar: 1 μm. Data information: In (**A–C**), data are presented as mean ± SEM. Two-tailed Student's t-test *p* value < 0.001 (\*\*\*) or non-significant (ns). Source data are available online for this figure.

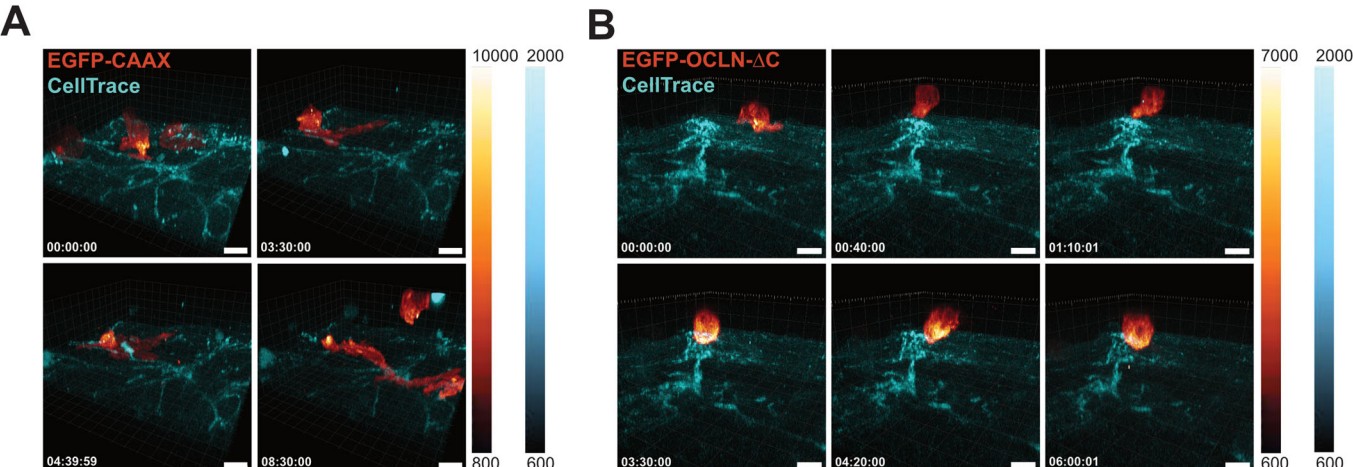

**Figure EV3. Monocytic OCLN transiently polarizes at monocyte-endothelial interaction sites during transmigration.**

(A, B) 3D time-lapse spinning disk confocal microscopy of transmigrating monocytes as in Fig. 1E, F. (A) Imaging of a primary monocyte transduced with EGFP-CAAX on hCMEC/D3 monolayer. Images were taken every 10 min. Scale bar: 10 μm. Full video can be found in Movie EV3. (B) Imaging of a primary monocyte transduced with EGFP-OCLN-ΔC on hCMEC/D3 monolayer. Images were taken every 10 min. Scale bar: 10 μm. Full video can be found in Movie EV4. Source data are available online for this figure.

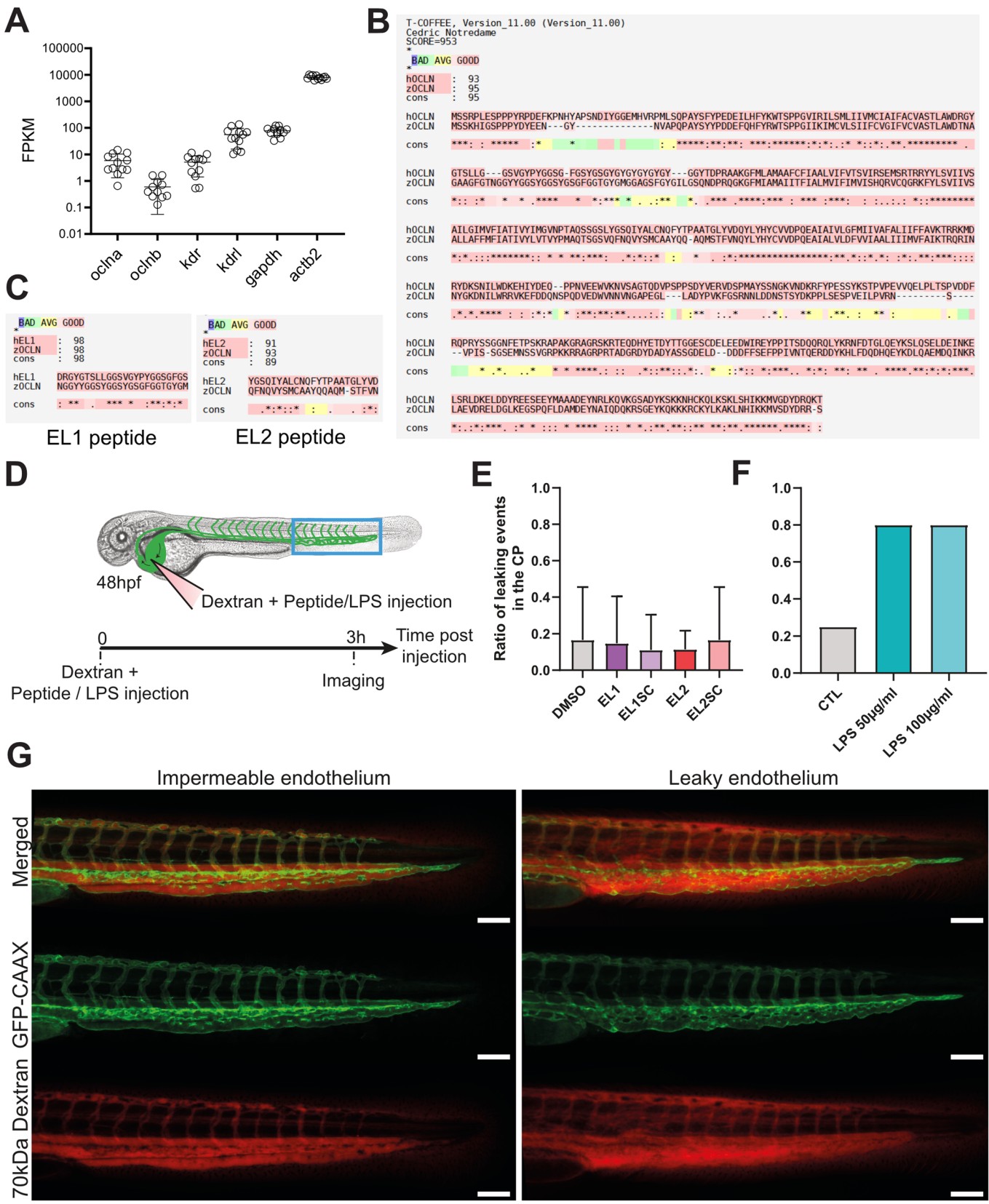

**Figure EV4.  Characterization of the zebrafish embryo model.**

(A) OCLN-expressing RNA for *oclna* and *oclnb* paralogues assessed in the endothelium of Tg(fli:egfp) zebrafish embryos from (Bonkhofer et al, 2019). Kdr, kdrl, *gapdh*, and *actb2* genes were used as housekeeping genes. (B) Comparative analysis of the human OCLN (hOCLN) and *danio rerio* (zebrafish) OCLN (zOCLN) amino acid sequences and consensus sequence using T-Coffee (see Methods for details). Red underlining of amino acids indicates good sequence similarity. (C) Comparative analysis of the EL1- and EL2-derived sequences of hOCLN with zOCLN. Red underlining of amino acids indicates good sequence similarity. (D) Schematic of the experimental procedure associated with the zebrafish model to test the effect of EL1, EL2 and their scramble peptides on endothelial permeability. Treatment with LPS is used as control for permeability inducing agents. (E) Bar graph of the ratio of leaky endothelium at 3 h post dextran and peptide injection. The experiment was carried out three independent times ($n_{DMSO} = 39$; $n_{EL1} = 25$; $n_{scEL1} = 26$; $n_{EL2} = 24$; $n_{scEL2} = 23$). (F) Bar graph of the ratio of leaky endothelium at 3 h post dextran and peptide injection. The experiment was carried out one time. ($n_{Control} = 12$; $n_{LPS-50µg/mL} = 15$; $n_{LPS-100µg/mL} = 15$). (G) Representative images of zebrafish embryos with an impermeable or leaky endothelium in the tail at 3 h post injection of PBS or LPS, respectively. Scale bar: 100 µm. Data information: In (A, D, E), data are presented as mean ± SD.

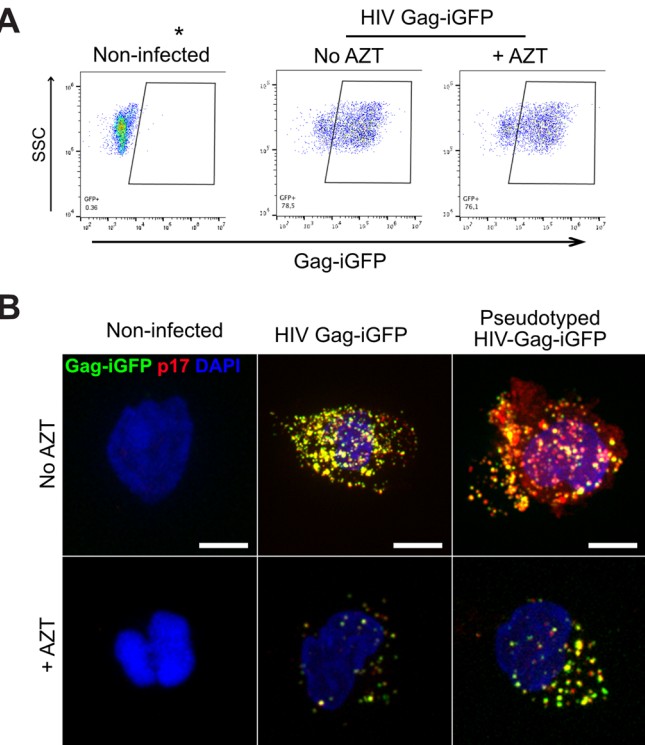

**Figure EV5. Characterization of primary monocyte infection by HIV-1 and transmigration.**

(A) Primary monocytes were non-infected or infected with HIV-1 (NLAD8) at MOI 1 for 48 h in the presence or absence of 10 μM AZT. The dot plots show the percentage of Gag-iGFP-expressing cells as a function of the side scatter (SSC) measurement as determined by flow cytometry. The data highlights that despite AZT treatment, monocytes are positive for Gag-iGFP as they carry fluorescent particles. (B) Primary monocytes were non-infected, infected with HIV-1 Gag-iGFP, or HIV-1 Gag-iGFP pseudotyped with VSV-G, at MOI 1 for 48 h in the presence or absence of AZT. Cells were fixed and stained for Gag p17 (red) and DAPI (blue) and Gag-iGFP is shown in green. Scale bar: 5 μm.

