## [Peer Review File · EMBO Reports]

Targeting monocytic Occludin impairs transendothelial migration and HIV neuroinvasion

Diana Brychka, Nilda Vanesa Ayala-Nunez, Amandine Dupas, Yonis Bare, Emma Partiot, Vincent Mittelheisser, Vincent Lucansky, Jacky Goetz, Nael Osmani, and Raphael Gaudin

Corresponding author(s): Raphael Gaudin (raphael.gaudin@irim.cnrs.fr)

Review Timeline:

Submission Date:	12th Oct 23
Editorial Decision:	14th Nov 23
Revision Received:	30th Mar 24
Editorial Decision:	2nd May 24
Revision Received:	11th Jun 24
Accepted:	17th Jun 24

Editor: Achim Breiling

Transaction Report:

Dear Dr. Gaudin,

Thank you for the submission of your research manuscript to EMBO reports. I have now received the reports from the three referees that were asked to evaluate your study, which can be found at the end of this email.

As you will see, the referees think that the findings are of interest. However, they have several comments, concerns, and suggestions, indicating that a major revision of the manuscript is necessary to allow publication of the study in EMBO reports. As the reports are below, and all the referee concerns need to be addressed, I will not detail them here.

Given the constructive referee comments, I would like to invite you to revise your manuscript with the understanding that all referee concerns must be addressed in the revised manuscript or in a detailed point-by-point response. Acceptance of your manuscript will depend on a positive outcome of a second round of review. It is EMBO reports policy to allow a single round of revision only and acceptance of the manuscript will therefore depend on the completeness of your responses included in the next, final version of the manuscript.

1) a .docx formatted version of the final manuscript text (including legends for main figures, EV figures and tables), but without the figures included. Figure legends should be compiled at the end of the manuscript text.

2) individual production quality figure files as .eps, .tif, .jpg (one file per figure), of main figures (up to 8) and EV figures. Please upload these as separate, individual files upon re-submission.

4) a complete author checklist, which you can download from our author guidelines

(<https://www.embopress.org/page/journal/14693178/authorguide>). Please insert page numbers in the checklist to indicate where the requested information can be found in the manuscript. The completed author checklist will also be part of the RPF.

5) that primary datasets produced in this study (e.g. RNA-seq, ChIP-seq, structural and array data) are deposited in an

appropriate public database. If no primary datasets have been deposited, please also state this in a dedicated section (e.g. 'No primary datasets have been generated and deposited'), see below.

The accession numbers and database should be listed in a formal "Data Availability" section (placed after Materials & Methods) that follows the model below. This is now mandatory (like the COI statement). Please note that the Data Availability Section is restricted to new primary data that are part of this study. This section is mandatory. As indicated above, if no primary datasets have been deposited, please state this in this section

Data availability

8) Regarding data quantification and statistics, please make sure that the number "n" for how many independent experiments were performed, their nature (biological versus technical replicates), the bars and error bars (e.g. SEM, SD) and the test used to calculate p-values is indicated in the respective figure legends (also for potential EV figures and all those in the final Appendix). Please also check that all the p-values are explained in the legend, and that these fit to those shown in the figure. Please provide statistical testing where applicable. Please avoid the phrase 'independent experiment', but clearly state if these were biological or technical replicates. Please also indicate (e.g. with n.s.) if testing was performed, but the differences are not significant. In case n=2, please show the data as separate datapoints without error bars and statistics. See also: <http://www.embopress.org/page/journal/14693178/authorguide#statisticalanalysis>

9) Please add scale bars of similar style and thickness to all the microscopic images, using clearly visible black or white bars (depending on the background). Please place these in the lower right corner of the images themselves. Please do not write on or near the bars in the image but define the size in the respective figure legend.

10) Please also note our reference format:

12) We now use CRedit to specify the contributions of each author in the journal submission system. CRedit replaces the author contribution section. Please use the free text box to provide more detailed descriptions and do not provide your final manuscript text file with an author contributions section. See also our guide to authors: <https://www.embopress.org/page/journal/14693178/authorguide#authorshipguidelines>

13) We would encourage you to use 'Structured Methods', our new Materials and Methods format. According to this format, the

Materials and Methods section should include a Reagents and Tools Table (listing key reagents, experimental models, software and relevant equipment and including their sources and relevant identifiers) followed by a Methods and Protocols section in which we encourage the authors to describe their methods using a step-by-step protocol format with bullet points, to facilitate the adoption of the methodologies across labs. More information on how to adhere to this format as well as downloadable templates (.doc or .xls) for the Reagents and Tools Table can be found in our author guidelines (section 'Structured Methods'):

14) Please order the manuscript sections like this, using these names:

Title page - Abstract - Keywords - Introduction - Results - Discussion - Materials and Methods - Data availability section - Acknowledgements - Disclosure and Competing Interests Statement - References - Figure legends - Expanded View Figure legends

I look forward to seeing a revised version of your manuscript when it is ready. Please let me know if you have questions or comments regarding the revision.

Yours sincerely,

Referee #1:

The manuscript by Brychka et al. investigates the role of human monocytic Occludin (hmOCLN) during neuroinvasion across the neurovascular unit (NVU) and its relevance during neuroinvasion by HIV.

The authors used RNAseq databases to identify tight junction associated proteins (TJAPs) expressed in human primary monocytes and in THP-1 cells. Deletion and overexpression of hmOCLN in monocytes and THP-1 cells pointed to participation of hmOCLN during transmigration of monocytes across a blood-brain barrier (BBB) in vitro model (hCMECV/D3). Immunofluorescence analyses showed that hmOCLN clustered at the monocyte-endothelium interface during transmigration. Peptides against the extracellular loops of hmOCLN inhibited transmigration of monocytes across hCMEC/D3 in vitro and in vivo in a zebrafish model. These peptides also prevented infiltration of monocytes and replication of HIV in a cortical organoid assay.

The key finding of this manuscript is identification of the importance of hmOCLN during neuroinvasion by monocytes that, to my knowledge, has not been reported before. This is a significant finding and of general interest to the molecular biology community. The performed experiments are well performed and presented and support the conclusions drawn.

I have the following points of concern:

1.) In Suppl. Figure S1B the authors show Western blots of THP-1 OCLN KO cells and THP-1 cells overexpressing EGFP-OCLN. In subsequent experiments the authors used THP-1 OCLN KO cells rescued with either EGFP-CAAX, EGFP-OCLN or EGFP-deltaC-OCLN (e.g. Figure 1C). Are the EGFP-fusion proteins expressed at comparable levels in the OCLN KO cells and can the authors show Western blots of these rescued cells that have been used for the experiments? Similarly, are the different fusion proteins expressed at similar levels in the primary monocytes (e.g. used in Figure 1D) as well?

Is the antibody against OCLN recognizing the deleted C-terminus of OCLN and can, therefore, not be used for detection of deltaC-OCLN? Would it be possible to use an antibody against EGFP instead?

2.) The authors determine permeability of hCMEC/D3 monolayers after overnight exposure to the OCLN-peptides using a lucifer yellow permeability assay. Has the integrity of the hCMEC/D3 monolayers also been analyzed during the transmigration assays using lucifer yellow or measurement of TEER (when possible)?

3.) The experiment shown in Figure 4E-F should be described in more detail. In the Materials & Methods the authors write that "after 24 h of incubation the top and bottom chambers were lysed for RNA extraction and RT-qPCR" (page 16, lines 464-465). In the discussion they write that in this assay "EL1 or EL2 peptide treatments of transmigrating monocytes exposed to HIV resulted in more than ten time less HIV RNA detectable in cerebral organoids" (page 11, lines 327-329). Was the whole material in the bottom compartment (including transmigrated monocytes) lysed? If so, is it clear that the RNA detected in the bottom compartment is located in the organoid (or to which extent is it located in the organoid)? Could the organoid be separated from the remainder of the material in the bottom compartment and both fractions be analyzed independently? Also, how exactly was the AZT treatment performed (added to all compartments, etc.)? If active viral replication is activated in the organoid, shouldn't have AZT an effect in this assay, when analyzing the bottom compartment?

4.) As the authors describe in the discussion, the peptide "EL1 did not show a significant inhibition of monocyte transmigration per se" (page 11, lines 329-330), but when analyzing HIV RNA in cerebral organoids following transmigration of monocytes (Figure 4) "the EL1 peptide also exhibited a strong effect" (page 11, line 330). The authors propose that "the OCLN-derived peptides act on various steps to inhibit HIV neuroinvasion" (Page 11, lines 330-331). The authors should discuss in more detail, which steps could be involved.

5.) I cannot find a description or a source for the EGFP-expression constructs (EGFP-CAAX, EGFP-OCLN, EGFP-deltaC-OCLN).

6.) Under Materials & Methods in the Statistics section only Student's t tests are mentioned, but at least in Figure 3 a Kruskal-Wallis test with Dunn's post hoc test is used.

7.) It is a little bit unfortunate, that the nomenclature of the TJAP-coding mRNAs is partly different between Figure 1A and Figure 1B, since this makes it difficult to identify the genes that have been knocked down (in case one is not absolutely familiar with the gene names).

Typos:

- page 3, lines 63-64: mechanisms ... remains > mechanisms ... remain
- page 3, line 66: Claudine-5 > Claudin-5
- page 4, lines 105-106: the 21 most expressed... > maybe better: 21 of the most expressed...
- page 8, line 224: differentiated cortical organoids ... incubated it > differentiated cortical organoids ... incubated them
- page 9, line 256: confirmed the present > confirmed the presence ?
- page 17, line 497: "Briefly, 100ug/ml 3 kDa Dextran was added to..." > end of the sentence missing
- page 27, line 785: ...were with CellTrace,... > ...were labeled with CellTrace,... ?
- page 29, line 806: what is meant with "upper right panel", also the blue arrow in the second panel from the left is not explained.

Referee #2:

In this paper, Brychka and colleagues tackle the molecular mechanisms that allow monocytes to cross the blood-brain barrier (BBB). They identify occluding (OCLN), a tight junction protein expressed by both endothelium and monocytes, as a significant player in this transmigration, and they test OCLN-derived peptides as potential blockers of this event, to protect from Trojan horse-mediated viral neuroinvasion.

Although promising and relying on advanced experimental methods, these data do not strongly support the author's conclusion, and will need to be reinforced before publication.

Major issues:

Time-lapse imaging of OCLN-EGFP overexpressing monocytes is exciting but statistical information is lacking to draw conclusions. How many crossing events were imaged? For how many cells followed? This is particularly important for negative controls.

The inhibitory activity of the scrEL2 peptide is a very serious issue. I do not understand the proposed explanation (line 180): "potentially because negatively charged amino acids of the EL2 sequence were partially conserved". According to the EL2 sequence on Fig S4B, this peptide contains only one negatively charged AA at PH7: its C-terminal aspartate. Even more problematic is the complete lack of control peptides for the rest of the study. These must be included in all experiments.

The zebrafish genome contains two ocln paralogues: oclna and oclnb. Both should be included on figure S4. Another issue lies with OCLN expression by zebrafish endothelial cells, which is not supported by the currently available data (such as ISH data on www.zfin.org), as far as I can tell. Please provide evidence for this expression, or discuss why OCLN should promote transmigration if no interaction between endothelial and monocytic OCLN can occur. This would also provide a trivial explanation as why these peptides do not impact vascular permeability of embryos (discussion line 300)

Transmigration experiments to organoids on figure 4: these are clearly difficult experiments, but the very high variance observed with DMSO controls makes it difficult to draw meaningful conclusions. Did the data pass normality tests allowing the use of t-tests? Anyway, as mentioned, control peptides are lacking here.

I am not convinced that monocytes inside organoids are productively infected with HIV - the mCherry signal on fig 4D looks punctiform and not cytosolic.

Minor issues:

Nomenclature is not consistent between panels 1B and 1C (eg tricellulin in 1A is named MARVELD2 on 1B, THP2 vs ZO-2, JAMs have letters in 1A and numbers on 1B...)

Besides occludin, a significant reduction in transmigration is seen by knocking down Claudin 23 on panel 1B. Obviously this paper is focused on OCLN, but this should be briefly discussed.

Line 54: transmigration of monocytes is said to occur by "squeezing between endothelial cells" - however in addition to this paracellular pathway, transcellular migration may also occur. This is mentioned in the discussion line 285 but should be mentioned from the beginning since there is no clear evidence that one or the other occur in the experiments reported here.

Please provide the ocln sequence mutations in the OCLN-deficient THP-1 clone.

Line 121 please explain why you chose to delete the C-terminal tail. I'd suggest to show the deleted region with a distinct color on panel 2A.

Fig S2C: rab7 distribution seems to be modified by OCLN overexpression. Is that the case?

The LPS-induced vascular leakiness control (fig S4) is not described in methods.

Fig 5C, top right panel; why is the p17 staining diffuse while GFP signal is not?

Referee #3:

This is an interesting study on the impact of monocytic occludin (ocln) on transendothelial monocyte migration. The findings are novel and should generate interest in the field. Microscopy study and presented movies are technical strengths of the paper. The use of peptides targeting the extracellular loops of ocln is another strength.

I have the following major comments:

1. What was the efficiency of silencing for individual TJ proteins shown in Fig 1B? Was it the same for all TJ proteins? If not (as this usually is the case), how these results can be compared to each other? What is the reason for huge variations in transmigration after some (but not other) TJ were silenced?
2. hCMEC/D3 cells do not generate tight monolayers as shown in numerous studies in the literature. Thus, transendothelial migration via hCMEC/D3 monolayers is not representative of primary cells. This is serious limitation of the study. Indeed, the authors demonstrate the monocytes are "squeezing" between endothelial cells, while, in fact, monocyte infiltration can also occur via a transcellular pathway.
3. Is an increase in transendothelial migration related to increased adhesion?
4. Ocln was recently shown to have several metabolic functions, including glucose metabolism (PMCID: PMC5951017) and even influencing HIV infection (PMCID: PMC4750406). Do these mechanisms play a role in the events studied in the present paper? Certainly, cell bioenergetics is important for cell migration.

Minor:

Claudin and not "claudine" on page 3.

What is the color coding on Figs 1C and 1D?

Referee #1:

The manuscript by Brychka et al. investigates the role of human monocytic Occludin (hmOCLN) during neuroinvasion across the neurovascular unit (NVU) and its relevance during neuroinvasion by HIV.

The authors used RNAseq databases to identify tight junction associated proteins (TJAPs) expressed in human primary monocytes and in THP-1 cells. Deletion and overexpression of hmOCLN in monocytes and THP-1 cells pointed to participation of hmOCLN during transmigration of monocytes across a blood-brain barrier (BBB) in vitro model (hCMECV/D3). Immunofluorescence analyses showed that hmOCLN clustered at the monocyte-endothelium interface during transmigration. Peptides against the extracellular loops of hmOCLN inhibited transmigration of monocytes across hCMEC/D3 in vitro and in vivo in a zebrafish model. These peptides also prevented infiltration of monocytes and replication of HIV in a cortical organoid assay.

The key finding of this manuscript is identification of the importance of hmOCLN during neuroinvasion by monocytes that, to my knowledge, has not been reported before. This is a significant finding and of general interest to the molecular biology community. The performed experiments are well performed and presented and support the conclusions drawn.

We thank the reviewer for acknowledging the significance of our findings and pointing out that our experiments were well performed and presented

I have the following points of concern:

1.) In Suppl. Figure S1B the authors show Western blots of THP-1 OCLN KO cells and THP-1 cells overexpressing EGFP-OCLN. In subsequent experiments the authors used THP-1 OCLN KO cells rescued with either EGFP-CAAX, EGFP-OCLN or EGFP-deltaC-OCLN (e.g. Figure 1C). Are the EGFP-fusion proteins expressed at comparable levels in the OCLN KO cells and can the authors show Western blots of these rescued cells that have been used for the experiments? Similarly, are the different fusion proteins expressed at similar levels in the primary monocytes (e.g. used in Figure 1D) as well? Is the antibody against OCLN recognizing the deleted C-terminus of OCLN and can, therefore, not be used for detection of deltaC-OCLN? Would it be possible to use an antibody against EGFP instead?

We thank the reviewer for these rightful comments. We are now showing in new **Figure EV1C** the expression levels of the EGFP-fused CAAX, EGFP-OCLN and EGFP-OCLN- Δ C in THP-1 cells by Western blot. Although we used the same promoter and plasmid backbone, the CAAX protein was expressed at higher levels, while the OCLN- Δ C was expressed at slightly lower levels, than WT OCLN. However, the interpretation of this blot needs to be done with caution, as proteins of higher molecular weight usually transfer less efficiently on membranes, and thus, this is not truly representative of protein expression levels. To be more unbiased, we also analysed the mean fluorescence intensities (MFI) of the EGFP-positive transduced THP1 cells and primary monocytes by flow cytometry and observed again a higher expression of the EGFP-CAAX construct compared to the others (**Figure EV1D,1H**). It is difficult to control expression levels of EGFP-fusion proteins in THP-1 and even more in primary monocytes. However, the fact that only EGFP-OCLN expression exhibits differential effect on transmigration in **Figure 1C and 1F**, we suggest that protein expression level is not a key component

altering the interpretation of our findings. To be transparent toward readers, we have however included this information and discussed it in the revised manuscript.

Our antibody does indeed recognize the C-ter of OCLN. Thus, it cannot be used to detect OCLN- Δ C. As suggested, we have now performed EGFP immunoblotting, allowing us to detect OCLN- Δ C at the expected size (new **Figure EV1C**).

2.) The authors determine permeability of hCMEC/D3 monolayers after overnight exposure to the OCLN-peptides using a lucifer yellow permeability assay. Has the integrity of the hCMEC/D3 monolayers also been analyzed during the transmigration assays using lucifer yellow or measurement of TEER (when possible)?

We thank the reviewer for his/her comment. We might have not been clear enough in the description of **Figure 2B** that the reviewer is referring to. The endothelium was incubated with peptides overnight, and lucifer yellow was performed for 2 h at the end of the incubation, while the peptides were not washed away. Thus, the LY permeability assay was somehow performed both at the end, and during peptide incubation. We believe that this protocol allows to see any potential peptide-induced permeability. However, none of the peptides induced significant changes of endothelial permeability despite the co-incubation of LY and peptides for 2 h. We have now modified the Result section to make these protocol details clearer for readers.

3.) The experiment shown in Figure 4E-F should be described in more detail. In the Materials & Methods the authors write that "after 24 h of incubation the top and bottom chambers were lysed for RNA extraction and RT-qPCR" (page 16, lines 464-465). In the discussion they write that in this assay "EL1 or EL2 peptide treatments of transmigrating monocytes exposed to HIV resulted in more than ten time less HIV RNA detectable in cerebral organoids" (page 11, lines 327-329). Was the whole material in the bottom compartment (including transmigrated monocytes) lysed? If so, is it clear that the RNA detected in the bottom compartment is located in the organoid (or to which extend is it located in the organoid)? Could the organoid be separated from the remainder of the material in the bottom compartment and both fractions be analyzed independently?

We apologize for the unclear explanation of this assay. In the previous version of the manuscript, we did analyse a mixture of supernatant and organoid from the bottom chamber. As requested, we repeated the whole experiment in order to test separately the supernatant and the organoid itself. In this experiment, presented in the new **Figure 4G-H**, we now show separately that the EL1/EL2 peptides significantly decrease HIV RNA both in the supernatant and in the organoid, highlighting the global effect of the peptide on transmigration. This data and the methodology have been now described in greater details in the revised version of the manuscript.

Also, how exactly was the AZT treatment performed (added to all compartments, etc.)? If active viral replication is activated in the organoid, shouldn't have AZT an effect in this assay, when analyzing the bottom compartment?

This is a very important point. AZT was added to the bottom compartment only, to mimic physiological situations, where ART would be concentrated in the bloodstream in much higher concentration than in the brain.

We agree with the reviewer that one may expect to see AZT decreasing HIV RNA levels. Here, we are claiming that the EL2 peptide can prevent HIV neuroinvasion by preventing the transmigration of monocytes carrying HIV particles. Indeed, we showed in **Figure 4A**, that HIV exposure enhances monocyte transmigration. This process is critical for HIV to cross an endothelium, as we now show in new **Figure 4B** that cell-free virus is basically unable to go through the endothelial monolayer without a monocytic carrier. In such big picture, productive infection is not a major point of concern to evaluate HIV neuroinvasion *per se*, at least in the first steps of this process.

Nevertheless, we showed in **Figure EV5** that monocytes uptake numerous incoming HIV particles, but were not productively infected at early stages. Hence, it is expected to find large amount of viral RNA in non-infected monocytes originating from incoming particle uptake. The addition of AZT in our assays did not show a difference between AZT-treated and non-treated conditions which can be interpreted as the fact that monocytes uptake similar amount of virus, which is independent of HIV replication, and thus AZT would play no role in this process. However, we acknowledge that while monocytes do not support very well HIV replication, monocytes can differentiate into macrophages/microglia in cerebral organoids, which are much more permissive to HIV replication. Actually, we can readily observe by videomicroscopy the increased appearance of HIV-Gag-imCherry protein. To make this observation clearer, we added snapshots of single channels in the **Figure 4C** of the revised manuscript and the quantification of the increase HIV fluorescence over time, indicative of productive viral infection. We also performed an experiment showing that HIV-infected monocyte-infiltrated cerebral organoids can exhibit HIV RNA levels that are decreased upon AZT treatment, as shown in the graph below. Unfortunately, we could perform this experiment only once (organoids being precious materials that take months to grow), and to retain thoroughness, we chose to present it here in the point-by-point but we did not include it in the main manuscript (but the point-by-point response shall be published together with the article). This observation was not made in the previous version of the manuscript because of two reasons: 1. One can assume that the amount of viral RNA produced by the monocyte-derived cells is hidden by the overwhelming amount of HIV RNA originating from non-replicative particle uptake, and 2. The time required to produce HIV RNA from myeloid cells is longer than for T cells for instance and thus, waiting longer allows us to observe more viral RNA, while waiting too long may result in monocyte death, and thus lower HIV RNA levels, making the analysis very tricky. Due to unclear interpretation of the AZT data, and to avoid confusion, we chose to remove the AZT data from the new **Figure 4G-H**.

4.) As the authors describe in the discussion, the peptide "EL1 did not show a significant inhibition of monocyte transmigration *per se*" (page 11, lines 329-330), but when analyzing HIV RNA in cerebral organoids following transmigration of monocytes (Figure 4) "the EL1 peptide also exhibited a strong effect" (page 11, line 330). The authors propose that "the OCLN-derived peptides act on various steps

to inhibit HIV neuroinvasion" (Page 11, lines 330-331). The authors should discuss in more detail, which steps could be involved.

As suggested by the reviewer here (and by reviewer #3 below), we have added in the discussion of the revised version of the manuscript additional hypothesis which could explain why the EL1 peptide has an effect on HIV RNA levels in organoids. We cannot exclude that HIV propagation in cerebral organoids necessitates host factors that the EL1 peptide modulates, nor that the peptides have direct antiviral activity, leading to an overall decrease of HIV RNA levels that is independent of transmigration.

5.) I cannot find a description or a source for the EGFP-expression constructs (EGFP-CAAX, EGFP-OCLN, EGFP-deltaC-OCLN).

Sorry for the missing information. This has now been added in the Method section.

6.) Under Materials & Methods in the Statistics section only Student's t tests are mentioned, but at least in Figure 3 a Kruskal-Wallis test with Dunn's post hoc test is used.

This has been corrected.

7.) It is a little bit unfortunate, that the nomenclature of the TJAP-coding mRNAs is partly different between Figure 1A and Figure 1B, since this makes it difficult to identify the genes that have been knocked down (in case one is not absolutely familiar with the gene names).

According to the reviewer's suggestion, we have now changed homogenized the nomenclature between the two panels.

Typos:

- page 3, lines 63-64: mechanisms ... remains > mechanisms ... remain

Corrected.

- page 3, line 66: Claudine-5 > Claudin-5

Corrected.

- page 4, lines 105-106: the 21 most expressed... > maybe better: 21 of the most expressed...

Corrected.

- page 8, line 224: differentiated cortical organoids ... incubated it > differentiated cortical organoids... incubated them

Corrected.

- page 9, line 256: confirmed the present > confirmed the presence ?

Corrected.

- page 17, line 497: "Briefly, 100ug/ml 3 kDa Dextran was added to..." > end of the sentence missing

We apologize for this missing end. This has been corrected.

- page 27, line 785: ...were with CellTrace,... > ...were labeled with CellTrace,... ?

Corrected.

- page 29, line 806: what is meant with "upper right panel", also the blue arrow in the second panel from the left is not explained.

We changed "upper right panel" to "second panel". The blue arrow has been removed.

Referee #2:

In this paper, Brychka and colleagues tackle the molecular mechanisms that allow monocytes to cross the blood-brain barrier (BBB). They identify occluding (OCLN), a tight junction protein expressed by both endothelium and monocytes, as a significant player in this transmigration, and they test OCLN-derived peptides as potential blockers of this event, to protect from Trojan horse-mediated viral neuroinvasion.

Although promising and relying on advanced experimental methods, these data do not strongly support the author's conclusion, and will need to be reinforced before publication.

We thank the reviewer for pointing out that our data are promising and we followed the reviewer's suggestions to reinforce our conclusions (see below).

Major issues:

Time-lapse imaging of OCLN-EGFP overexpressing monocytes is exciting but statistical information is lacking to draw conclusions. How many crossing events were imaged? For how many cells followed? This is particularly important for negative controls.

We agree with the reviewer that the imaging data are very interesting, and we tried very hard to obtain quantitative data to include in the manuscript, but as said in the previous version of the manuscript, "such imaging is very challenging and we could not reliably quantify and time this event, although we could reproducibly observe it". Nevertheless, in order to be more specific, we have manually quantified what we meant by "repeatedly observed", but because of the relative power of such quantitative analysis, we chose not to make a graph, but to write it in the main text. We believe that this info makes the data clearer for readers, giving a sense of the frequency of the observed events.

The inhibitory activity of the scrEL2 peptide is a very serious issue. I do not understand the proposed explanation (line 180): "potentially because negatively charged amino acids of the EL2 sequence were partially conserved". According to the EL2 sequence on Fig S4B, this peptide contains only one negatively charged AA at PH7: its C-terminal aspartate. Even more problematic is the complete lack of control peptides for the rest of the study. These must be included in all experiments.

We thank the reviewer for this very important comment. We removed the “negatively charged amino acid” statement in the revised manuscript. To try to understand the reason for the effect of the scrEL2 control peptide, we analysed their putative secondary structure (see below). We found that a helical structure was partially conserved in the two peptides (EL2 and scrEL2 a, H letter highlighting potential helical domains). Therefore, we purchased a new scramble (scrEL2 b) that better shuffled this helical domain. Additional experiments using the former scramble and new scramble confirmed that they had less inhibitory activity than the EL2 peptide, although both scrambles still exhibited some anti-transmigratory activity (see new **Figure 2C-D**).

EL2	scrEL2 a	scrEL2 b
YGSQIYALCNQFYTPAATGLYVD	PQAYDFTGNGSYLCTLYAYVIAQ	LIFYATQCYDATYVNGPLAQYSG
OrigSeq : YGSQIYALCNQFYTPAATGLYVD	OrigSeq : PQAYDFTGNGSYLCTLYAYVIAQ	OrigSeq : LIFYATQCYDATYVNGPLAQYSG
Jnet : ---HHHHHHH-----EEE--	Jnet : -----HHHHHHHHH--	Jnet : ---HHHH-----

As requested by the reviewer, we perform new experiments with transmigration of HIV-exposed monocytes toward organoids including these control peptides and confirmed that the EL2 peptide was significantly decreasing HIV neuroinvasion, while both scramble controls did not show significant effect (see new **Figure 4G-H**).

We believe that these new data are very important to reinforce our claim on the role of the EL2 peptide for the inhibition of monocyte transmigration and HIV neuroinvasion, and thank the reviewer for this thoughtful remark. Of note, the additional experiments with organoids were very demanding in terms of time and expenses, and we could not engage at this stage in additional experiments in zebrafish. Although we agree that a negative control in zebrafish is of interest, we still show the absence of significant effect of the EL1 peptide in zebrafish, and overall, the message remain unchanged, i.e. the EL2 peptide inhibits monocyte transmigration *in vivo*. We sincerely hope that the efforts we made to improve significantly this study will satisfy the reviewer.

The zebrafish genome contains two ocln paralogues: oclna and oclnb. Both should be included on figure S4. Another issue lies with OCLN expression by zebrafish endothelial cells, which is not supported by the currently available data (such as ISH data on www.zfin.org), as far as I can tell. Please provide evidence for this expression, or discuss why OCLN should promote transmigration if no interaction between endothelial and monocytic OCLN can occur. This would also provide a trivial explanation as why these peptides do not impact vascular permeability of embryos (discussion line 300).

Following reviewer’s comments, we took advantage of our previously published RNAseq dataset performed on zebrafish endothelia (Follain G et al, Sci Rep, 2021) to evaluate the expression levels of the two OCLN isoforms as well as fibronectin and control genes. We found that oclnb was barely expressed while oclna, the one we used for subsequent analyses, was expressed 10 to 100 times more in the zebrafish endothelium. This is indeed an important information that we have now included in new **Figure EV4A** and in the text of the revised version of the manuscript.

Transmigration experiments to organoids on figure 4: these are clearly difficult experiments, but the very high variance observed with DMSO controls makes it difficult to draw meaningful conclusions. Did the data pass normality tests allowing the use of t-tests? Anyway, as mentioned, control peptides are lacking here.

We thank the reviewer for this comment. The mentioned experiment has been fully repeated adding more repeats, and additional controls as requested (see new **Figure 4G-H**). Indeed, relatively high heterogeneity is observed in organoids, which can be explained by the fact that each organoid grows in its own way, and although we used sized-matched organoids as much as possible, we cannot fully control the specific organoid composition in each condition. Moreover, we further separated organoids from the supernatant of the lower chamber to measure HIV RNA levels in both conditions. This additional analysis further reinforced the statement that HIV was less neuroinvasive upon peptide treatment. Here the addition of numerous organoids for each condition made the data more robust. Moreover, as suggested by the reviewer, we included in this experiment control scramble peptides, one scrEL1 and 2 scrEL2, and found that none of them had significant activity against HIV neuroinvasion. These new data are now presented in the revised version of **Figure 4G-H**.

I am not convinced that monocytes inside organoids are productively infected with HIV - the mCherry signal on fig 4D looks punctiform and not cytosolic.

As mentioned above to reviewer 1, we agree with the reviewer that one may expect to see AZT decreasing HIV RNA levels. Here, we are claiming that the EL2 peptide can prevent HIV neuroinvasion by preventing the transmigration of monocytes carrying HIV particles. Indeed, we showed in **Figure 4A**, that HIV exposure enhances monocyte transmigration. This process is critical for HIV to cross an endothelium, as we now show in new **Figure 4B** that cell-free virus is basically unable to go through the endothelial monolayer without a monocytic carrier. In such big picture, productive infection is not a major point of concern to evaluate HIV neuroinvasion *per se*, at least in the first steps of this process.

Nevertheless, we showed in **Figure EV5** that monocytes uptake numerous incoming HIV particles, but were not productively infected at early stages. Hence, it is expected to find large amount of viral RNA in non-infected monocytes originating from incoming particle uptake. The addition of AZT in our assays did not show a difference between AZT-treated and non-treated conditions which can be interpreted as the fact that monocytes uptake similar amount of virus, which is independent of HIV replication, and thus AZT would play no role in this process. However, we acknowledge that while monocytes do not support very well HIV replication, monocytes can differentiate into macrophages/microglia in cerebral organoids, which are much more permissive to HIV replication. Actually, we can readily observe by videomicroscopy the increased appearance of HIV-Gag-imCherry protein. To make this observation clearer, we added snapshots of single channels in the **Figure 4C** of the revised manuscript and the quantification of the increase HIV fluorescence over time, indicative of productive viral infection. We also performed an experiment showing that HIV-infected monocyte-infiltrated cerebral organoids can exhibit HIV RNA levels that are decreased upon AZT treatment, as shown in the graph above. Due to unclear interpretation of the AZT data at this stage, and to avoid confusion, we chose to remove the AZT data from the new **Figure 4G-H**.

Minor issues:

Nomenclature is not consistent between panels 1B and 1C (eg tricellulin in 1A is named MARVELD2 on 1B, THP2 vs ZO-2, JAMs have letters in 1A and numbers on 1B...)

This has now been homogenized.

Besides occludin, a significant reduction in transmigration is seen by knocking down Claudin 23 on panel 1B. Obviously this paper is focused on OCLN, but this should be briefly discussed.

It is true that CLDN23 is also significantly decreasing transmigration, but the phenotype being less prominent, and literature being scarce, we chose to focus on OCLN. We agree with the reviewer that the data on CLDN23 remains of interest and we have now discussed briefly this observation.

Line 54: transmigration of monocytes is said to occur by "squeezing between endothelial cells" - however in addition to this paracellular pathway, transcellular migration may also occur. This is mentioned in the discussion line 285 but should be mentioned from the beginning since there is no clear evidence that one or the other occur in the experiments reported here.

As requested by the reviewer, we have now mentioned transcellular migration in the beginning in the introduction.

Please provide the ocln sequence mutations in the OCLN-deficient THP-1 clone.

Although clonal, the THP-1 cells have unknown number of chromosomes, and thus, two or more alleles. The CRISPR-induced mutation on each allele would differ in nature and lengths, giving sequencing data difficult to interpret, as the mix population of PCR products (at least 2) would be co-sequenced. Here we are presenting the outcome of the OCLN KO at the protein level, which we believe is also highly relevant for the characterization of these cells, while genome sequencing provides indirect information and .

Line 121 please explain why you chose to delete the C-terminal tail. I'd suggest to show the deleted region with a distinct color on panel 2A.

As suggested, we have now explained our choice for C-ter deletion in the revised version of our manuscript.

Fig S2C: rab7 distribution seems to be modified by OCLN overexpression. Is that the case?

This is a very good observation indeed. Actually, it seems that it is rather the CAAX overexpression that is associated to some increased Rab7 expression and aggregation, while Rab7 is usually lowly expressed, as in the EGFP-Rab7 condition. We show in the new **Figure EV1C** that EGFP-CAAX construct is strongly overexpressed in monocytes compared to EGFP-OCLN, and thus, the cells might increase its degradative properties (through Rab7/lysosomes). However, the EGFP-OCLN-DC can also be used as a control, expressed at similar levels that of EGFP-OCLN. We preferred not to comment on this as the interpretation remains mostly speculative.

The LPS-induced vascular leakiness control (fig S4) is not described in methods.

This information has been added.

Fig 5C, top right panel; why is the p17 staining diffuse while GFP signal is not?

This might be due to image contrasts, but there is some GFP signals in the cytosol their too. The contrasts were the same for all images and increasing contrasts for this image may saturate gfp signal in the others and thus we preferred to keep them as is.

Referee #3:

This is an interesting study on the impact of monocytic occludin (ocln) on transendothelial monocyte migration. The findings are novel and should generate interest in the field. Microscopy study and presented movies are technical strengths of the paper. The use of peptides targeting the extracellular loops of ocln is another strength.

We thank the reviewer for highlighting the strengths of our study.

I have the following major comments:

1. What was the efficiency of silencing for individual TJ proteins shown in Fig 1B? Was it the same for all TJ proteins? If not (as this usually is the case), how these results can be compared to each other? What is the reason for huge variations in transmigration after some (but not other) TJ were silenced?

The siRNA screening was optimized by testing electroporation conditions to knock-down a control mRNA with > 70% efficiency in THP-1 cells. We chose optimal conditions described in the Methods to perform the screen, but could not check knock-down efficiency on all target genes. Because we chose a pool of four siRNA per gene, we tend to believe that good knock-down efficiency can be achieved. However, we acknowledge the fact that our dataset does not fully rule out the potential involvement of other TJAPs during monocyte transmigration, and we have included this statement in the discussion of the revised manuscript.

The reasons observed for the huge variation is the fact that some TJAP KD had no effect in one experiment, but decreased/increased in another. Because of the lack of reproducibility, we did not further focus on these TJAPs.

2. hCMEC/D3 cells do not generate tight monolayers as shown in numerous studies in the literature. Thus, transendothelial migration via hCMEC/D3 monolayers is not representative of primary cells. This is serious limitation of the study.

We thank the reviewer for this comment. The D3 cells have tighter junctions than most other cell lines (see Weksler B et al, Fluids Barriers CNS, 2013), but indeed, their TJs do not reach the strength seen with primary cells, more complex multicellular models, nor in vivo BBB. To better inform readers, we have now added a paragraph in the discussion to acknowledge those limitations.

Indeed, the authors demonstrate the monocytes are "squeezing" between endothelial cells, while, in fact, monocyte infiltration can also occur via a transcellular pathway.

As also suggested by the reviewer #2, we have now explained since the Introduction section the two possible transmigration routes. Moreover, the discussion specifically mentions that it remains unclear whether OCLN could be involved in transcellular migration, for the complete reader's information.

3. Is an increase in transendothelial migration related to increased adhesion?

We thank the reviewer for this rightful comment. Indeed, we previously showed that the EL2 peptide does not modulate monocyte adhesion, but we did not check specifically whether the increase of transmigration seen in OCLN overexpressed cells was due to increased adhesion on the endothelial layer specifically. There, we now show in new **Figure 1D-E** that primary monocyte adhesion onto fibronectin and endothelial cells was not significantly affected by OCLN expression levels, confirming that monocytic OCLN is unlikely to play a role in monocyte adhesion.

4. Ocln was recently shown to have several metabolic functions, including glucose metabolism (PMCID: PMC5951017) and even influencing HIV infection (PMCID: PMC4750406). Do these mechanisms play a role in the events studied in the present paper? Certainly, cell bioenergetics is important for cell migration.

We thank the reviewer for pointing out these articles. Although we did not prove it, one can hypothesize that EL1 and/or EL2 exhibit direct antiviral activity against HIV, and these studies could provide some mechanistic insights to explain it. This information has been included in the revised version of the manuscript to enrich the discussion, citing these two references.

Minor:

Claudin and not "claudine" on page 3.

Corrected

What is the color coding on Figs 1C and 1D?

Each symbol's color corresponds to an individual experiment (n = 4) for Fig 1C and to an individual blood donor (n = 4) in Fig 1D. This info has now been added in the legend.

Dear Dr. Gaudin,

Thank you for the submission of your revised manuscript to our editorial offices. I have already forwarded to you the reports I have received from the three referees that I asked to re-evaluate the study, you will find again below. As you know, referees #1 and #2 support the publication of the study in EMBO reports, although they have some further points and suggestions to improve the manuscript. Referee #3 remains more critical and indicates remaining concerns. I now went through your rebuttal letter and ask you to address the remaining points in a final revised manuscript accordingly. Please also provide a final p-b-p-response with your resubmission, addressing the remaining points of the referees.

- We plan to publish your manuscript in the Report format (as also indicated by you in the submission system), as there are not more than 5 main and EV figures. For a Scientific Report we require that results and discussion sections are combined in a single chapter called "Results & Discussion". Please do this for your manuscript. For more details, please refer to our guide to authors:

<http://www.embopress.org/page/journal/14693178/authorguide#researcharticleguide>

- Please make sure that the number "n" for how many independent experiments were performed, their nature (biological versus technical replicates), the bars and error bars (e.g. SEM, SD) and the test used to calculate p-values is indicated in the respective figure legends (also for potential EV figures and all those in the final Appendix). Please also check that all the p-values are explained in the legend, and that these fit to those shown in the figure. Please provide statistical testing where applicable. Please avoid the phrase 'independent experiment', but clearly state if these were biological or technical replicates. Please also indicate (e.g. with n.s.) if testing was performed, but the differences are not significant. In case n=2, please show the data as separate datapoints without error bars and statistics. See also:

<http://www.embopress.org/page/journal/14693178/authorguide#statisticalanalysis>

If $n < 5$, please show single datapoints for diagrams. Presently, some diagrams seem to show only partial or no statistics. Please check. Moreover:

- Please define the annotated p values ***/** in the legend of figure 4b; EV 2c; as appropriate.

- Please indicate the statistical test used for data analysis in the legends of figures 4b; EV 2a-c.

- Please note that in figure 3d; there is a mismatch between the annotated p values in the figure legend and the annotated p values in the figure file that should be corrected.

- Please note that information related to n is missing in the legends of figures 2b; EV 2a-c.

- Please note that the error bars are not defined in the legends of figures 2b; 4b; EV 2a-c; EV 4a.

- Please add to each legend (main and EV figures, where applicable) a 'Data Information' section explaining the statistics used or providing information regarding replicates and scales. See:

- Please add scale bars of similar style and thickness to all microscopic images, using clearly visible black or white bars (depending on the background). Please place these in the lower right corner of the images themselves. Please do not write on or near the bars in the image but define the size in the respective figure legend. Presently, the scale bars are rather varied, some with text nearby, some too thin. Please provide more uniform scale bars without text.

- Please use our reference format:

- We now use CRediT to specify the contributions of each author in the journal submission system. CRediT replaces the author contribution section. Please use the free text box to provide more detailed descriptions. Thus, please do not provide your final manuscript text file with an author contributions section (also not as part of the acknowledgements). See also guide to authors: <https://www.embopress.org/page/journal/14693178/authorguide#authorshippinguidelines>

- Please make sure that all the funding information is also entered into the online submission system and is complete and similar to the one in the manuscript text file (in the Acknowledgements). Presently, the grants 'ANRS MIE, the French embassy in Slovakia and the Program Hubert Curien (PHC) Stefanik, University of Strasbourg and ARC (Association pour la Recherche contre le Cancer), INSERM and University of Strasbourg, La Ligue contre le Cancer' are only mentioned in the Acknowledgements.

- All the EV figures have incorrect labels in the files: "Suppl. Figure S1" instead of "Figure EV1". Please either correct this or remove these labels from the EV figure files.

- In the manuscript text (page 9) a panel 4I is called out (Figure 4H-I), but no such panel exists. Please check.
- Please name the materials and methods section just 'Methods'.
- The "Data Availability" section should only list primary datasets created in the study that have been deposited to a database or external repository. If no datasets have been deposited, please state here: 'No primary datasets have been generated or deposited'.
- The entire nomenclature for the movie files needs to be corrected. The source file names, file titles, legends, and callouts need correction to "Movie EVx". The legends need to be removed from the manuscript text file and each should be zipped up together with its corresponding movie file as a README.txt file. Moreover, there are 2 files 'Movie S5' uploaded. Please check.
- Please name the antibodies table 'Table 1' and add a legend. Or upload this information (maybe together with primer, peptide or chemicals information) as reagents and tools table. I have attached templates for that in word or excel format. Please upload the filled in table to the manuscript tracking system as 'Reagent Table' file. Please also add callouts to this table to the methods section. The example linked below shows how the table will display in the published article and includes examples of the type of information that should be provided for the different categories of reagents and tools. Please list your reagents/tools using the categories provided in the template and do not add additional subheadings to the table. Reagents/tools that do not fit in any of the specific categories can be listed under "Other":
https://www.embopress.org/pb%2Dassets/embo-site/msb_177951_sample_FINAL.pdf
- You indicate in the author checklist that a dual use research restriction select agent has been used. Dual Use Research of Concern (DURC) is life sciences research that, based on current understanding, can be reasonably anticipated to provide knowledge, information, products, or technologies that could be directly misapplied to pose a significant threat with broad potential consequences to public health and safety, agricultural crops and other plants, animals, the environment, materiel, or national security. What agent did you sue and where is that described? Please check. If you really think that your study could fall under dual use research restriction, we need a detailed explanation for this to decide if we proceed with publication. See also: <http://www.embopress.org/page/journal/14693178/authorguide#biosecurity>
- Please add a paragraph titled 'Biosafety' to the methods section gathering all information on where and how biosafety-relevant experiments with viruses were performed and that these were approved, and by whom (institution, government).
- Thanks for providing the requested source data (SD). However, the SD need to be re-grouped. All SD for one figure needs to be ZIPed up together and then one folder per figure should be uploaded. SD for EV figures need to be renamed accordingly and can be uploaded ZIPed together in one folder. The request for SD for Fig. 4F referred to the V1 version. Please make sure that the corresponding SD is now provided, and also any SD for new main figure panels (not present in the originally submitted version) are uploaded.

In addition, I would need from you:

Best,

Referee #1:

In the revised version of the manuscript the authors have answered to all my comments and significantly improved the manuscript. I have only minor points to address:

- I have some problems understanding how the graphs in Figures 1D, 1E and 1F were generated. If the data were normalized to EGFP-CAAX of each experiment, why are the EGFP-CAAX values not all located at 100% (as in Figure 1C)? Maybe the

authors could explain in a little more detail. Also, red dashes representing the mean of experiments combined together are only shown for Figures 1C and 1F (not mentioned in the legend to Figure 1F), but not for Figures 1D and 1E.

- In the legend to Figure 4F-H it is mentioned that the experiments were performed in presence or absence of AZT and that no AZT was added to the bottom chamber, but as I understand the AZT data are not shown in Figure 4 anymore. This figure legend might need to be corrected.

-Typos:

- page 3, line 76: human deficiency virus (HIV) > the abbreviation HIV was already used in line 74
- page 4 lines 112-113: while Claudin-23 > while silencing of Claudin-23
- page 5, line 136: increased compare to > increased compared to
- page 10, lines 299-300: cells are have tighter TJs > cells have tighter TJs
- page 12, line 376: associated to increase expression > associated to increased expression
- page 26, line 855: CellTrace marker (cyan) added the addition of the monocytes prior > added prior to the addition of the monocytes (?)
- page 26, lines 855-856: Timescales are shown is the bottom right corner and corresponds > Timescales are shown in the bottom right corner and correspond

Referee #2:

Bruchka and colleagues have addressed most of my requests, but there are a few remaining points.

Major requests

The authors have added a figure showing the expression of *oclna*, *oclnb* in zebrafish endothelium that they say come from their previously published RNAseq data of zebrafish endothelia in (Follain et al. 2012). However, when I check this paper, I find a RNAseq dataset from HUVEC cells only. FACS purification of zebrafish endothelium was made but only qRT-PCR is reported. It appears that you would have to fully describe the generation of this RNAseq data in this manuscript. (Alternatively you may use other data sources, as a search with the terms "danio" and "endothelium" turns out 29 hits on the BioProject database at NCBI/

The absence of control peptides in the zebrafish experiments remains a major problem. I understand that these experiments are not at the core of the work and that they are not trivial to perform, so I will not require additional experiments. However, I will ask the authors to acknowledge this issue upfront and tone down their text accordingly (eg, line 218, replace "highlights" by "suggests" and equivalent changes in discussion). In particular, do not state in the abstract that transmigration is significantly inhibited in zebrafish embryos.

Minor points

Line 209-213: the appropriate term for zebrafish *oclna* and *oclnb* is "paralogues", not "isoforms" - they are different genes.

Line 244: "quixotic" is a cool word but I suspect that it would be more suitable here to write "exquisite"

Referee #3:

The authors improved the manuscript; however, several of my concerns remain:

Item 1. The efficiency of silencing for individual TJ proteins shown in Fig 1B.

The authors responded that the siRNA screening was optimized by testing electroporation conditions to knock-down a control mRNA with > 70% efficiency in THP-1 cells. This approach is not satisfactory. It is really hard to understand why the authors could not check the efficiency of silencing protocol for the TJ genes on Fig 1B. What is the reason that this could not be checked? There are 21 genes on this graph. Importantly, other experiments are built on the findings from Figure 1. Could the authors check efficiency of transfection at least for selected genes?

Item 2. The authors explained that "the reasons observed for the huge variation is the fact that some TJAP KD had no effect in one experiment, but decreased/increased in another." This explanation is also unacceptable. The lack of data reproducibility is a major concern, which is also reflected on several other Figures (e.g., Fig 1C, E, F, Fig 2D, etc). This high variability is difficult to understand.

Item 3. The authors responded: "The D3 cells have tighter junctions than most other cell lines (see Weksler B et al, Fluids Barriers CNS, 2013), but indeed, their TJs do not reach the strength seen with primary cells, more complex multicellular models, nor in vivo BBB. To better inform readers, we have now added a paragraph in the discussion to acknowledge those limitations." This explanation is unacceptable. My comment warranted control experiments on primary cells and not a paragraph discussing limitations.

Point-by-point response

Referee #1:

In the revised version of the manuscript the authors have answered to all my comments and significantly improved the manuscript. I have only minor points to address:

- I have some problems understanding how the graphs in Figures 1D, 1E and 1F were generated. If the data were normalized to EGFP-CAAX of each experiment, why are the EGFP-CAAX values not all located at 100% (as in Figure 1C)? Maybe the authors could explain in a little more detail. Also, red dashes representing the mean of experiments combined together are only shown for Figures 1C and 1F (not mentioned in the legend to Figure 1F), but not for Figures 1D and 1E.

We apologize for the lack of clarity. The data were normalized to the average of EGFP-CAAX obtained from all experiments. As one can see in Figure 1F, the red bar for EGFP-CAAX is at 100% as it represents the mean of all the individual experiments for EGFP-CAAX. We chose not to had the mean for panel 1D and 1E to avoid overcrowding the graph, but one can appreciate that the datapoints for the EGFP-CAAX conditions are found around the 100% normalization. We added further clarification in the legend section.

- In the legend to Figure 4F-H it is mentioned that the experiments were performed in presence or absence of AZT and that no AZT was added to the bottom chamber, but as I understand the AZT data are not shown in Figure 4 anymore. This figure legend might need to be corrected.

This is a mistake from the previous version. It has now been corrected.

-Typos:

- page 3, line 76: human deficiency virus (HIV) > the abbreviation HIV was already used in line 74.

Corrected

- page 4 lines 112-113: while Claudin-23 > while silencing of Claudin-23

Corrected

- page 5, line 136: increased compare to > increased compared to

Corrected

- page 10, lines 299-300: cells are have tighter TJs > cells have tighter TJs

Corrected

- page 12, line 376: associated to increase expression > associated to increased expression

Corrected

- page 26, line 855: CellTrace marker (cyan) added the addition of the monocytes prior > added prior to the addition of the monocytes (?)

Corrected

- page 26, lines 855-856: Timescales are shown is the bottom right corner and corresponds > Timescales are shown in the bottom right corner and correspond

Corrected

Referee #2:

Bruchka and colleagues have addressed most of my requests, but there are a few remaining points.

Major requests

*The authors have added a figure showing the expression of *oclna*, *oclnb* in zebrafish endothelium that they say come from their previously published RNAseq data of zebrafish endothelia in (Follain et al. 2012). However, when I check this paper, I find a RNAseq dataset from HUVEC cells only. FACS purification of zebrafish endothelium was made but only qRT-PCR is reported. It appears that you would have to fully describe the generation of this RNAseq data in this manuscript. (Alternatively you may use other data sources, as a search with the terms "danio" and "endothelium" turns out 29 hits on the BioProject database at NCBI)/*

We thank the reviewer for pointing this out. Indeed, the whole dataset was not published in the cited article and we apologize for this mistake. The dataset will be published in a manuscript currently in preparation. Therefore, as suggested by the reviewer, we analysed instead an existing transcriptomic dataset from Bonkhofer F et al., Nature Comm, 2010 and replaced the graph from Figure EV4A with this new analysis. The data showed the same trend of *oclna* being expressed significantly more than *oclnb*, hence, not changing the initial message.

The absence of control peptides in the zebrafish experiments remains a major problem. I understand that these experiments are not at the core of the work and that they are not trivial to perform, so I will not require additional experiments. However, I will ask the authors to acknowledge this issue upfront and tone down their text accordingly (eg, line 218, replace "highlights" by "suggests" and equivalent changes in discussion). In particular, do not state in the abstract that transmigration is significantly inhibited in zebrafish embryos.

As per the reviewer's request, we have toned-down the claims in the text, and specifically, we changed "highlights" by "suggested" and "significantly" was removed in the abstract.

Minor points

*Line 209-213: the appropriate term for zebrafish *oclna* and *oclnb* is "paralogues", not "isoforms" - they are different genes.*

Corrected

Line 244: "quixotic" is a cool word but I suspect that it would be more suitable here to write "exquisite"

Corrected

Referee #3:

The authors improved the manuscript; however, several of my concerns remain: Item 1. The efficiency of silencing for individual TJ proteins shown in Fig 1B. The authors responded that the siRNA screening was optimized by testing electroporation conditions to knock-down a control mRNA with > 70% efficiency in THP-1 cells. This approach is not satisfactory. It is really hard to understand why the authors could not check the efficiency of silencing protocol for the TJ genes on Fig 1B. What is the reason that this could not be checked? There are 21 genes on this graph. Importantly, other experiments are built on the findings from Figure 1. Could the authors check efficiency of transfection at least for selected genes?

We thank the reviewer for his/her comment. Unfortunately, the screen was performed in 2017 and the RNAs were not kept. Buying new siRNA, new primers and perform the whole screen all over again would be lengthy, expensive, and we are currently unable to do it. Adding this information would change the message of our manuscript but we agree that may have missed genes from our list that play a role in monocyte transmigration. Hence, we have now acknowledged this limitation in the revised version of the manuscript.

Item 2. The authors explained that "the reasons observed for the huge variation is the fact that some TJAP KD had no effect in one experiment, but decreased/increased in another." This explanation is also unacceptable. The lack of data reproducibility is a major concern, which is also reflected on several other Figures (e.g., Fig 1C, E, F, Fig 2D, etc). This high variability is difficult to understand.

We understand the concern of the reviewer, and would like to point out that we used mostly primary monocytes isolated from donors, which exhibit extensive differences to start with, and it is often observed that primary monocytes have quite heterogeneous phenotypes. Of note however, our data on OCLN in the screen were very reproducible. To be transparent on variability in Figure 1B, we are now showing each datapoint.

Item 3. The authors responded: "The D3 cells have tighter junctions than most other cell lines (see Weksler B et al, Fluids Barriers CNS, 2013), but indeed, their TJs do not reach the strength seen with primary cells, more complex multicellular models, nor in vivo BBB. To better inform readers, we have now added a paragraph in the discussion to acknowledge those limitations." This explanation is unacceptable. My comment warranted control experiments on primary cells and not a paragraph discussing limitations.

We apologize for the inconvenience. Unfortunately, we do not have access nor have the expertise in the lab to isolate and culture ex vivo primary endothelia of blood-brain barrier origin. The hCMEC/D3 cells have been reported in > 600 articles, and they are not as physiological as primary BBB-derived endothelia indeed, so we clearly stated this limitation in accordance with the previous reviewer's comment. We also believe that the confirmation of the efficiency of the OCLN peptide in vivo (and thus in primary cells somehow), confirming the importance of our finding in more physiological context.

Dr. Raphael Gaudin
CNRS
IRIM
1919 route de Mende
Montpellier 34293
France

Dear Dr. Gaudin,

I am very pleased to accept your manuscript for publication in the next available issue of EMBO reports. Thank you for your contribution to our journal.

Yours sincerely,
